# Melting and fragmentation laws from the evolution of two large Southern Ocean icebergs estimated from satellite data

N. Bouhier[1], J. Tournadre[1], F. Rémy[2], and R. Gourves-Cousin[1]

[1]Laboratoire d'Océanographie Physique et Spatiale, IFREMER, Université Bretagne-Loire, Plouzané, France
[2]Laboratoire d'Etudes en Géophysique et Océanographie Spatiales, UMR 5566 | CNES - CNRS , Toulouse, France

**Correspondence:** Jean Tournadre
(Jean Tournadre@ifremer.fr)

**Abstract.** The evolution of the thickness and area of two large Southern Ocean icebergs that have drifted in open water for more than a year is estimated through the combined analysis of altimeter data and visible satellite images. The observed thickness evolution is compared with iceberg melting predictions from two commonly used melting formulations, allowing us to test their validity for large icebergs. The first formulation, based on a fluid dynamics approach, tends to underestimate basal melt rates, while the second formulation, which considers the thermodynamic budget, appears more consistent with observations. Fragmentation is more important than melting for the decay of large icebergs. Despite its importance, fragmentation remains poorly documented. The correlation between the observed volume loss of our two icebergs and environmental parameters highlights factors most likely to promote fragmentation. Using this information, a bulk model of fragmentation is established that depends on ocean temperature and iceberg velocity. The model is effective at reproducing observed volume variations. The size distribution of the calved pieces is estimated using both altimeter data and visible images and is found to be consistent with previous results and typical of brittle fragmentation processes. These results are valuable in accounting for the freshwater flux constrained by large icebergs in models.

## 1 Introduction

According to recent studies (Silva et al., 2006; Tournadre et al., 2015, 2016), most of the total volume of ice (~60%) calved from the Antarctic continent is transported into the Southern Ocean by large icebergs (i.e. >18km in length). However, their basal melting, that is of the order of $320 \, \text{km}^3 \, \text{yr}^{-1}$, accounts for less than 20% of their mass loss, and the majority of ice loss ($1.500 \, \text{km}^3 \, \text{yr}^{-1}$ ~80%) is achieved through breaking into smaller icebergs (Tournadre et al., 2016). Large icebergs actually act as a reservoir to transport ice away from the Antarctic coastline into the ocean interior, while fragmentation can be viewed as a diffusive process. It generates plumes of small icebergs that melt far more efficiently than larger ones and whose geographical distribution constrains the input into the ocean.

Global ocean models that include iceberg components (Gladstone et al., 2001; Jongma et al., 2009; Martin and Adcroft, 2010; Marsh et al., 2015; Merino et al., 2016) show that basal ice-shelf and iceberg melting have different effects on the ocean circulation. Numerical model runs with and without icebergs show that the inclusion of icebergs in a fully coupled general circulation model (GCM) results in significant changes in the modelled ocean circulation and sea-ice conditions around Antarctica

(Jongma et al., 2009; Martin and Adcroft, 2010; Merino et al., 2016). The transport of ice away from the coast by icebergs and the associated freshwater flux cause these changes (Jongma et al., 2009). Although the results of these modelling studies are not always in agreement in terms of ocean circulation or sea ice extent they all highlight the important role that icebergs play in the climate system, and they also show that models that do not include an iceberg component are effectively introducing
systematic biases (Martin and Adcroft, 2010).

However, despite these modelling efforts, the current generation of iceberg models are not yet able to represent the full range of iceberg sizes observed in nature from growlers ( $\leq 10$ m) to "giant" tabular icebergs ( $\geq 10$ km).

The iceberg size distribution has also strong impact on both circulation and sea ice as shown by Stern et al. (2016). Furthermore, all current iceberg models fail in accounting for the size transfer of ice induced by fragmentation, as in these models
small icebergs cannot stem from the breaking of bigger ones.

The two main decay processes of icebergs, melting and fragmentation, are still quite poorly documented and not fully represented in numerical models. Although iceberg melting has been widely studied (Huppert and Josberger, 1980; Neshyba, 1980; Hamley and Budd, 1986; Jansen et al., 2007; Jacka and Giles, 2007; Helly et al., 2011), very few validations of melting laws have been published (Jansen et al., 2007), especially for large icebergs. Large uncertainties still remain for the melting
laws to be used in numerical models.

The calving of icebergs from glaciers and ice shelves has been quite well studied (e.g Holdsworth and Glynn (1978); Fricker et al. (2002); Benn et al. (2007); MacAyeal et al. (2006); Amundson and Truffer (2010)) and empirical calving laws have been proposed (Amundson and Truffer, 2010; Bassis, 2011). However, very few studies have been dedicated to the breaking of icebergs. Analysing the decay of Greenland icebergs, Savage (2001) proposed three distinct fragmentation mechanisms.
Firstly, flexural breakups by swell induced vibrations in the frequency range of the iceberg bobbing on water that could cause fatigue and fracture at weak spots (Goodman et al., 1980; Schwerdtfeger, 1980; Wadhams et al., 1983). Secondly, two mechanisms resulting from wave erosion at the waterline, calving of ice overhangs and buoyant footloose mechanism (Wagner et al., 2014). Scambos et al. (2008), using satellite images, ICESat altimeter and field measurements analysed the evolution of two Antarctic icebergs and identified three styles of calving during the drift: "rift calving", which corresponds to the calving of
large daughter icebergs by fracturing along preexisting flaws, "edge wasting", the calving of numerous small narrow icebergs and "rapid disintegration", which is characterised by the rapid calving of numerous icebergs.

The pieces calved from icebergs drift away from their parent under the action of wind and ocean currents as a function of size, shape and draft (Savage, 2001). This dispersion can create large plumes of icebergs that can represent a significant contribution to the freshwater flux over vast oceanic regions where no large icebergs are observed (Tournadre et al., 2016).
The size distribution of the calved pieces is needed to analyse and understand the transfer of ice between the different iceberg scales and thus to estimate the freshwater flux. It is also important for modelling purposes. Savage et al. (2000), using aerial images and in situ measurements, estimated the size distribution of small bergy bits (<20 m in length) calved from deteriorating Greenland icebergs. However, until now, no study has been published on the size distribution of icebergs calved from large Southern Ocean icebergs.

Recent progresses in satellite altimeter data analysis allow us to estimate the small (<3 km in length) iceberg distribution and volume as well as the freeboard elevation profile and volume of large icebergs (Tournadre et al., 2016).The location, area and volume of small icebergs from 1992 to present is contained in a database distributed by CERSAT, as well as monthly fields of probability of presence, mean area and volume of ice (Tournadre et al., 2016). It is now possible to estimate the thickness variations and thus the melting of large icebergs. A crude estimate of the large iceberg area is also available from the National Ice Center but it is not precise enough to allow analysisof the area lost by fragmentation. A more precise area analysis can be conducted by analysing satellite images such as those for the Moderate Resolution Imaging Spectro- radiometer (MODIS) onboard the Aqua and Terra satellites (Scambos et al., 2005).

Two large icebergs, B17a and C19a, which have drifted for more than one year in open water (see figure 1) away from other large icebergs and which have been very well sampled by altimeters and MODIS, have been selected to study the melting and fragmentation of large Southern Ocean tabular icebergs. Their freeboard evolution, and thus thickness, is estimated from satellite altimeter data, while their area and shape have been estimated from the analysis of MODIS images. The icebergs area and thickness evolution is then used to test the validity of the melting models used in iceberg numerical modelling and to analyse the fragmentation process. The two icebergs were also chosen because they have very different characteristics. While C19a was one of the largest iceberg on record (>1000 km$^2$) and drifted for more than 2 years in the South Pacific, B17a was relatively small (200 km$^2$) and drifted in the Weddell Sea. The large plumes of small icebergs generated by the decay of both large icebergs can be detected by altimeters and MODIS images. The ALTIBERG database and selected MODIS images can be used to analyse the size distribution of fragments.

The present paper is organised as follows. Section 2 describes the data used in the study, including the environmental parameters (such as ocean temperature, current speed, ..) necessary to estimate melting and fragmentation. Section 3 presents the evolution of the two selected icebergs. In section 4, the two melting laws widely used in the literature, forced convection and thermal turbulence exchange, are confronted with the observed melting of B17a and C19a. The final section analyses the fragmentation process and proposes a fragmentation law. It also investigates the size distribution of pieces calved from large icebergs.

## 2 Data

### 2.1 Iceberg Data

The National Ice Center (NIC) Southern Hemisphere Iceberg database contains the position and size (length and width) estimated by analysis of visible or SAR images of icebergs larger than 10 nautical miles (19 km) along at least one axis. It is updated weekly. Every iceberg is tracked, and when imagery is available, information is updated and posted. The Brigham Young University (BYU) Center for Remote Sensing maintains an Antarctic Iceberg Tracking Database for icebergs larger than 6 km in length (Stuart and Long, 2011). Using six different satellite scatterometer instruments, they produced an iceberg tracking database that includes icebergs identified in enhanced resolution scatterometer backscatter. The initial position for

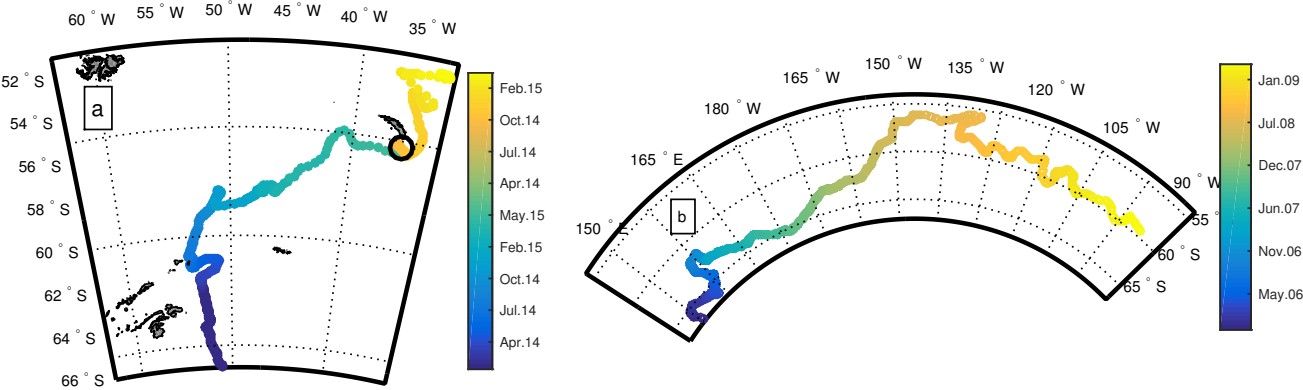

**Figure 1.** Trajectories of B17a (a) and C19a (b) icebergs. The black circle locates the B17a grounding site . The colorscale represents the time along the trajectory.

each iceberg is located based on a position reported by the NIC or by the sighting of a moving iceberg in a time series of scatterometer images.

In 2007, Tournadre (2007) demonstrated that any target emerging from the sea surface (such as an iceberg) can produce a detectable signature in HR altimeter wave forms. Their method enables us to detect icebergs in the open ocean only, and to estimate their area. Due to constraints on the method, only icebergs between 0.1km$^2$ and ~9 km$^2$ can be detected. Nine satellite altimetry missions have been processed to produce a 1992-present database of small iceberg locations, area, volume and mean backscatter (Tournadre et al., 2016). The monthly mean probability of presence, area and volume of ice over a regular polar (100x100 km$^2$) or geographical (1$^o$x2$^o$) grid are also available and are distributed on the CERSAT website.

Altimetry can also be used to measure the freeboard elevation profile of large icebergs (McIntyre and Cudlip, 1987; Tournadre et al., 2015). Combining iceberg tracks from NIC and the archives of three Ku band altimeters, Jason-1, Jason-2 and Envisat, Tournadre et al. (2015) created a database of daily position, freeboard profile, length, width, area and volume of all the NIC/BYU large icebergs covering the 2002-2012 period. For example, B17a was sampled by 152 altimeter passes during its drift and C19a by 258 passes (see figure 2).

### 2.2 Visible Images

The weekly estimates of iceberg lengths and widths provided by NIC are manually estimated from satellite images and they are not accurate enough to precisely compute the iceberg area and its evolution. A careful re-analysis of the MODIS imagery from the Aqua and Terra satellites was thus conducted to precisely estimate the C19a and B17a area until their final detectable

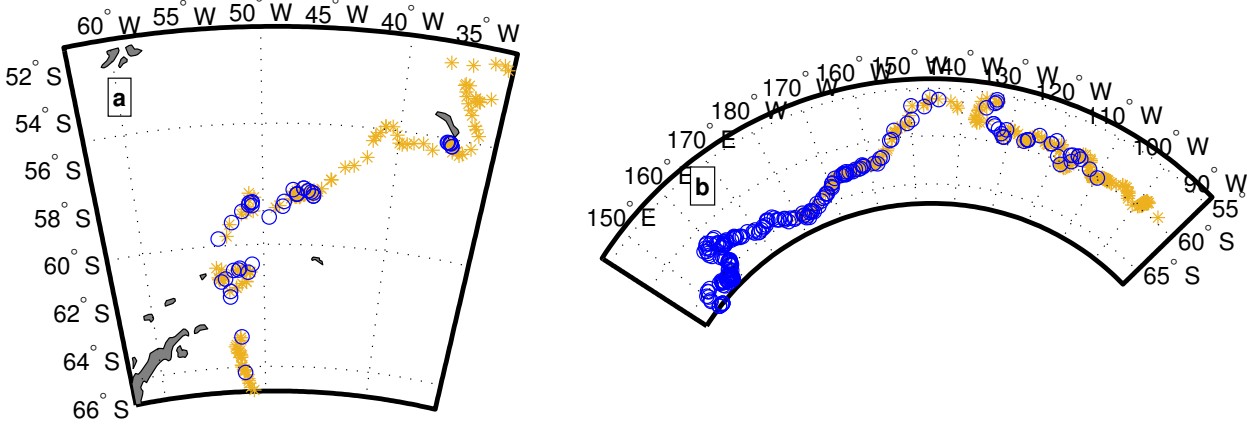

**Figure 2.** Sampling of B17a (a) and C19a (b) icebergs by MODIS (beige stars) and altimeters (blue circles).

collapse. The images have been systematically collocated with the two icebergs using the NIC/BYU track data. It should be noted that in some areas of high iceberg concentration, especially when B17a reaches the “iceberg alley”, NIC/BYU regularly mistakenly followed another iceberg, or lost its track when it became quite small. Here, more than 1500 images were collocated and selected. The level 1B calibrated radiances from the two higher resolution (250 m) channels (visible channels 1 and 2 at 645 and 860 nm frequencies) were used to estimate the iceberg’s characteristics. For each image with good cloud clover and light conditions, a supervised shape analysis was performed. Firstly, a threshold depending on the image light conditions is estimated and used to compute a binary image. The connected components of the binary image are then determined using standard Matlab© image processing tools and finally the iceberg’s properties, centroid position, major and minor axis lengths and area are estimated. On a number of occasions, the iceberg’s surface was obscured by clouds, but visual estimation was possible because the image contrast was sufficient to discern edges through clouds. For these instances, the iceberg’s edge and shape were manually estimated. The final analysis is based on 286 valid images for B17a, and 503 for C19a. The locations of the MODIS images for B17a and C19a are given in figure 2 while four examples of iceberg area estimates are given in figure 3. The comparison of area for consecutive images shows that the area precision is around 2-3%.

### 2.3 Environmental data

Several environmental parameters along the icebergs trajectories are also used in this study. Due to the lack of a better alternative, the sea surface temperature (SST) is used as a proxy for the water temperature. The difference between the SST and the temperature at the base of the iceberg will introduce an error in the melt rate computation as shown by Merino et al. (2016). Using results from an Ocean General Circulation Model, they also compared the mean SST and the average temperature over the first 150 m from the surface showing that the mean difference is less than 0.5°C for most of the Southern Ocean. The level-4 satellite analysis product ODYSSEA, distributed by the Group for High-Resolution Sea Surface Tempera-

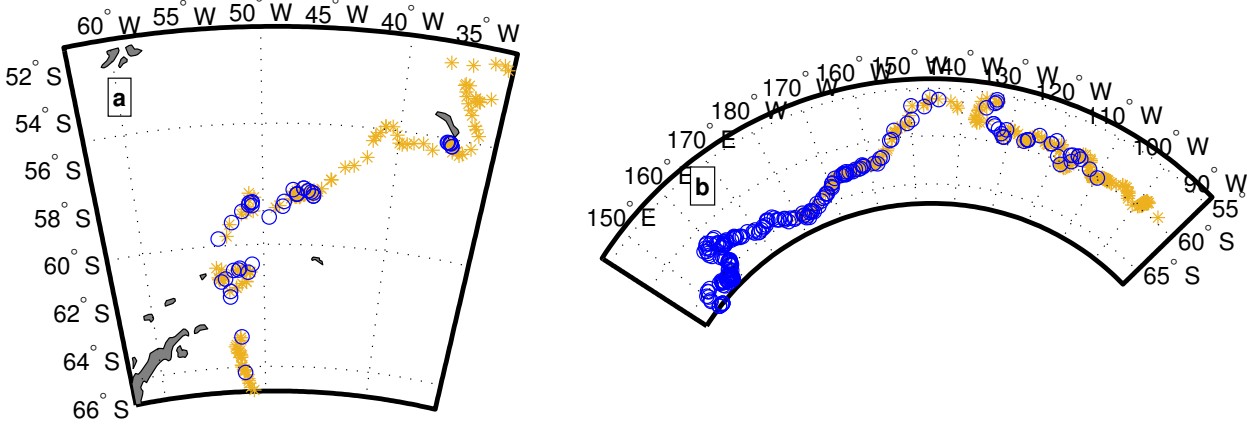

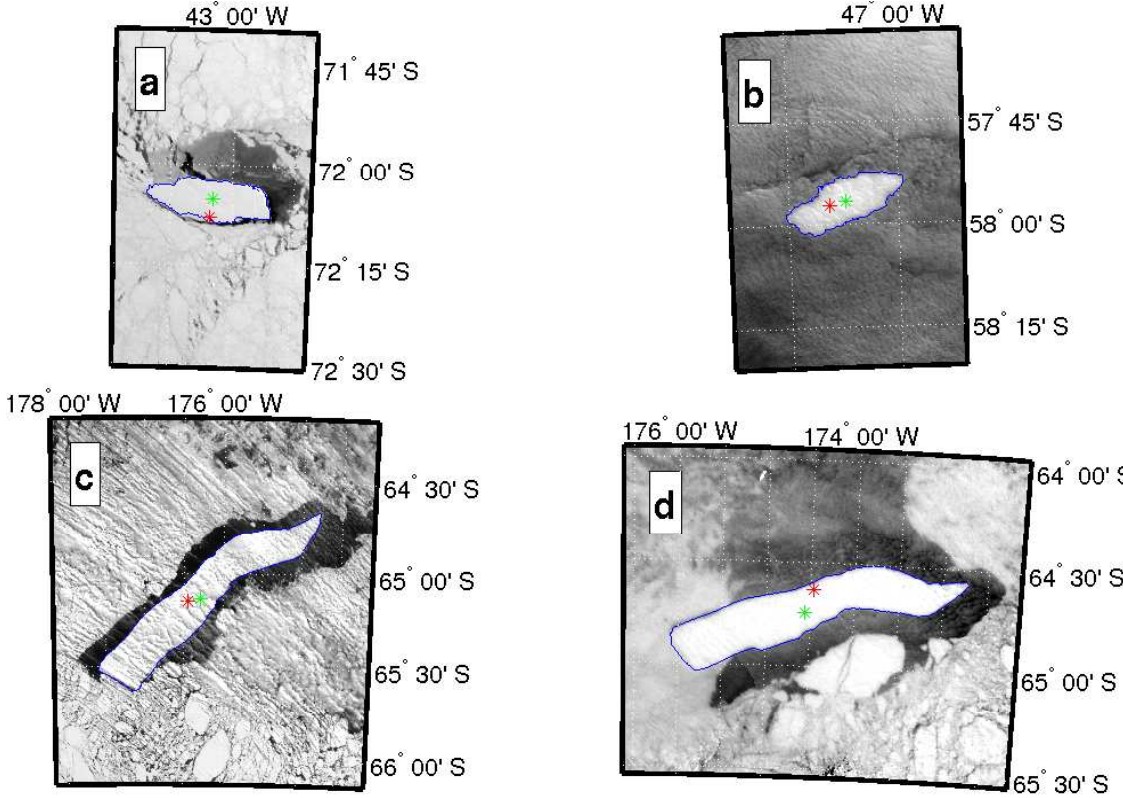

**Figure 3.** Example of B17a (a and b) and C19a (c and d) area estimate using Modis images. The blue lines represent the iceberg perimeter, the red and green crosses represent the NIC and MODIS iceberg's positions respectively.

ture (GHRSST) has been used. It is generated by merging infrared and microwave sensors and using optimal interpolation to produce daily cloud-free SST fields at 10 km resolution over the globe. The sea ice concentration data are from the CERSAT level-3 daily concentration product, available on a 12.5 km polar stereographic grid from the SSM/I radiometer observations. The wave height and wave peak frequencies come from the global Wave Watch3 hindcast products from the IOWAGA

5  project (http://wwz.ifremer.fr/iowaga/). The AVISO Maps of Absolute Dynamic Topography & absolute geostrophic velocities (MADT) provides a daily multi-mission absolute geostrophic current on a 0.25° regular grid that is used to estimate the current velocities at the iceberg locations.

## 3 Melting and fragmentation of B17a and C19a

### 3.1 B17a

Iceberg B17a originates from the breaking of giant tabular B17 near Cape Hudson in 2002. It then drifted for 10 years along the continental slope within the "coastal current", until it reached the Weddell Sea in summer 2012 (see figure 1-a). It travelled

within sea ice at a speed ranging from 2 to 12 cm.s$^{-1}$, coherent with previous observational studies (Schodlok et al., 2006). It crossed the Weddell Sea while drifting within sea ice and reached the open water in April 2014. It was then caught in the western branch of the Weddell Gyre and drifted north in the Scotia Sea until it grounded, in October 2014, near South Georgia, a common grounding spot for icebergs. It remained there for almost 6 months until it finally left its trap in March 2015 and drifted back northward until its final demise in early June 2015. B17a was a "medium size" big iceberg, with primary dimensions of 35

x 14 km$^2$ and an estimated freeboard of 52 m, resulting in an original volume of 113 km$^3$ and a corresponding mass of ~103 Gt. Before 2014, B17a freeboard and area remained almost constant while it drifted within sea ice. After March 2014, B17a started to drift in open water and to melt and break. During its drift in open water, from March 2014 to June 2015, B17a was sampled by 200 MODIS images and 41 altimeter passes. Figure 4-a presents the satellite freeboard and area measurements as well as the daily interpolated values. The standard deviation of freeboard estimate computed from the freeboard elevation profiles is

±3 m. The standard deviation of the iceberg area has been estimated by analysing the area difference between images taken the same day. It is of the order of 3-4%. During this drift in the Weddell Sea, it experienced different basal melting regimes: firstly, when it left the peninsula slope current, with negative SST's and low drift speeds (see figure 4-b and -d), it was subject to an average melt rate of 5.7m.month$^{-1}$; then it drifted more rapidly within the Scotia Sea and experienced a mean thickness decrease of 15 m.month$^{-1}$, and finally it melted at a rate close to 20 m.month$^{-1}$ as it accelerated its drift before its grounding.

As for fragmentation, the area loss was limited (40 km$^2$ in 250 days, i.e. less than 10%) but then accelerated as B17a became trapped (80 km$^2$ in 70 days). The area loss slowed down for the second half of the grounding, only to increase dramatically once B17a was released and before it collapsed a few days later. This could be related to an embrittlement of the iceberg structure, potentially under the action of unbalanced buoyancy forces while grounded (Venkatesh, 1986; Wagner et al., 2014; Stern et al., 2015).

The cumulative total volume loss, basal melting, breaking are presented in figure 4-e. These terms are computed from the mean thickness and area as follows: the basal melting volume loss $M$ at day $i$ is the sum of the products of iceberg surface, $S$ (in m$^2$), by the daily variation of thickness, $dT$

$$M(i) = \sum_{k=1}^{i} S(k)dT(k) \tag{1}$$

and the breaking loss $B$ (in m$^3$) is the sum of the products of thickness, $T$, by the daily variation of surface, $dS$

$$B(i) = \sum_{k=1}^{i} dS(k)T(k) \tag{2}$$

For large icebergs, the sidewall erosion/melting, which is of the order of some meters per day, can be considered negligible compared to breaking. As B17a started to drift in open water, its mass varied slowly at first, mainly through melting. Between January 2014 and March 2015, basal melting accounted for more than 60% of the total volume loss, whereas fragmentation was responsible for 30% of the loss. However, after November 2014 breaking became dominant as the iceberg started to break up more rapidly.

## 3.2  C19a

Our second iceberg of interest is the giant C19a which was one of the fragments resulting from the splitting of C19, the second largest tabular iceberg on record. C19a was born offshore Cap Adare (170°E) in 2003 and was originally oblong and narrow, around 165 km long and 32 km wide with an estimated freeboard of ~40 m, i.e. a volume of about 1000 km$^3$ and a mass of 900 Gt. It drifted mainly northeastward for almost 4 years, in sea ice for most of the time, until it first entered the open ocean in summer 2005 (see figure 1). It was temporarily re-trapped by the floes in winter 2006 and eventually left the ice coverage permanently in late spring 2007. It then drifted within the Antarctic Circumpolar Current and eventually close to the Polar Front and its warm waters until its final demise in April 2009 in the Bellingshausen Sea. Before November 2007, C19a experienced very little change except a very mild melting (not presented in the figure). Its volume was 880 km$^3$ ( ~790 Gt) in December 2007 when it finally entered the open sea. During its final drift, from December 2007 to March 2009, C19a was sampled by 317 MODIS images and 69 altimeter passes (see figure 2-b). The C19a area and freeboard are presented in figure 5 as well as SST, sea state and volume loss. While the volume loss was mainly due to melting before this date, breaking dominated afterwards. Basal melting only explains 25% of the total volume decrease (see figure 5-e). B17 thickness loss was almost 5 times faster than that of C19, the latter experiencing mean basal melt rates ranging from 1 m.month$^{-1}$ to 3 m.month$^{-1}$ in most of its drift (and as much as 13 m.month$^{-1}$ in its last month, which was characterised by very high water temperatures). As for fragmentation, its main volume loss mechanism (75%), its area loss was first mild while it progressed in colder waters (around 2.6 km$^2$.day$^{-1}$), and started to increase as soon as it entered positive temperature waters, with an average loss of 9.5 km$^2$.day$^{-1}$ and with dramatic shrinkages of 340 km² and 370  km² lost in 10 days that correspond to large fragmentation events.

## 4  Melting models

Apart from fragmentation, the basal melting of icebergs accounts for the largest part of the total mass loss (Martin and Adcroft, 2010; Tournadre et al., 2015). Although firn densification (see Appendix A1 for an estimate of the associated freeboard change) and surface melting can also contribute, it is the main cause of thickness decrease. It can be mainly attributed to the turbulent heat transfer arising from the difference of speed between the iceberg and the surrounding water. Two main approaches have been used to compute the melting rate and to model the evolution of iceberg and the freshwater flux (see for example Bigg et al. (1997); Gladstone et al. (2001); Silva et al. (2006); Jongma et al. (2009); Merino et al. (2016); Jansen et al. (2007)). The first one is based on the forced convection formulation proposed by Weeks and Campbell (1973), while the second one uses the thermodynamic formulation of Hellmer and Olbers (1989) and the turbulent exchange velocity at the ice-ocean boundary. The

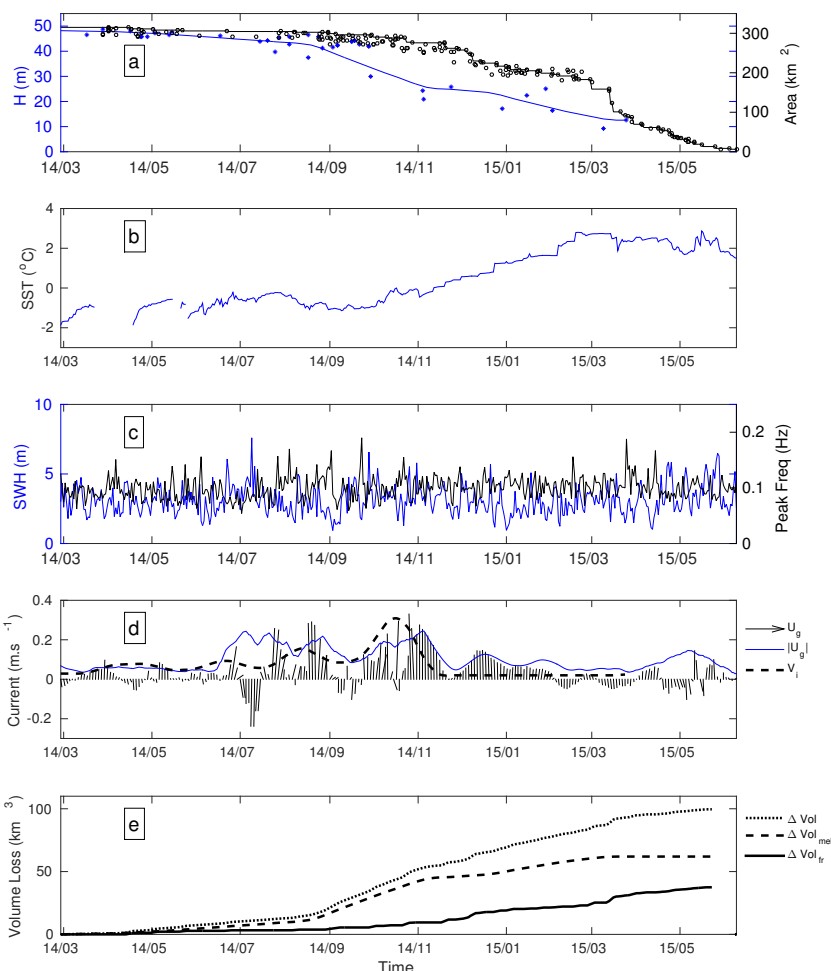

**Figure 4.** (a) B17a Area (in km$^2$) and freeboard (in m). The black and blue lines represent the interpolated daily area and freeboard and the black circles and blue crosses the MODIS area and altimeter freeboard estimates. (b) ODYSSEA Sea surface temperature (in °C). (c) Significant wave height in m (blue line) and peak frequency in Hz (green line). (d) AVISO geostrophic current (black arrows) and current velocity (blue line) and iceberg velocity (dashed black line). (e) Total volume loss (dotted line), volume loss by melting (dashed line) and by fragmentation (solid line).

first model has been exclusively used to compute iceberg basal melt rate while the second model has been primarily developed and used to estimate ice shelf melting. The B17a and C19a data sets allow us to confront these two formulations with melting

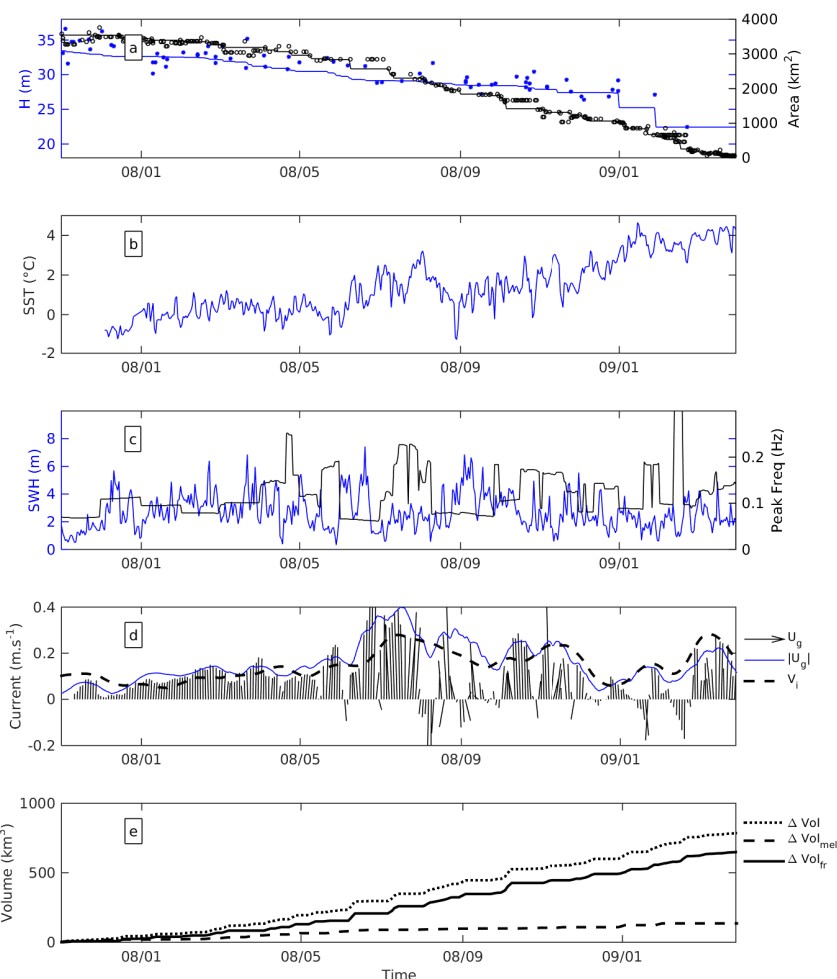

**Figure 5.** (a) C19a Area (in km$^2$) and freeboard (in m). The black and blue lines represent the interpolated daily area and freeboard and the black circles and blue crosses the MODIS area and altimeter freeboard estimates. (b) ODYSSEA Sea surface temperature (in °C). (c) Significant wave height in m (blue line) and peak frequency in Hz (green line). (d) AVISO geostrophic current (black arrows) and current velocity (blue line) and iceberg velocity (dashed black line). (e) Total volume loss (dotted line), volume loss by melting (dashed line) and by fragmentation (solid line). .

measurements for two icebergs of different shapes and sizes and under different environmental conditions and to test their validity for large icebergs.

## 4.1  Forced convection of Weeks and Campbell

The forced convection approach of Weeks and Campbell (1973) is based on the fluid mechanics formulation of the heat-transfer coefficient for a fully turbulent flow of fluid over a flat plate. The basal convective melt rate $M_b$ is a function of both temperature and velocity differences between the iceberg and the ocean. It is expressed (in m.day$^{-1}$) as (Gladstone et al., 2001; Bigg et al., 1997):

$$M_b = C|\overrightarrow{V_w} - \overrightarrow{V_i}|^{0.8}\frac{T_w - T_i}{L^{0.2}} \tag{3}$$

with $\overrightarrow{V_w}$ being the current speed (at the base of the iceberg), $\overrightarrow{V_i}$ the iceberg speed, $T_i$ and $T_w$ the iceberg and water temperature, $L$ the iceberg's length (longer axis) and $C = 0.58\text{K}^{-1}\text{m}^{0.4}\text{s}^{0.8}\text{day}^{-1}$. This expression has been widely used in numerical models (Bigg et al., 1997; Gladstone et al., 2001; Martin and Adcroft, 2010; Merino et al., 2016; Wagner et al., 2017). As water temperature at keel depth is not available, the sea surface temperature (SST) is used as a proxy. The SST for each iceberg is presented in figures 4-b and 5-b. The first unknown quantity in (3), the iceberg's temperature $T_i$ can be at the time of calving as low as -20°C (Diemand, 2001). Icebergs can sometimes drift for several years. During its travel the iceberg's surface temperature will depend on the ablation rate. When ablation is limited, i.e. in cold waters, the ice can theoretically warm up to 0°C, while in warmer waters the rapid disappearance of the outer layers tends to leave colder ice near the surface. The surface ice temperature could thus theoretically vary from -20°C to 0°C but is commonly taken at -4°C (Løset, 1993; Martin and Adcroft, 2010; Gladstone et al., 2001).

The mean daily iceberg speed can be easily estimated from the iceberg track. Numerical ocean circulation models are not precise enough to provide realistic current speed in this region. The comparison of iceberg velocities and AVISO geostrophic currents presented in figures 4-d and 5-d shows that the iceberg velocity is sometimes significantly larger than the AVISO velocities. They are thus not reliable enough to compute the melt rate. $V_w$ is thus treated as unknown.

The basal melt is computed using Equation 3 for $V_w$ from 0 to 3 m.s$^{-1}$ by 0.01 steps and $T_i$ from -20 to 2°C by 0.1°C steps. The positive temperatures are used to test the model's convergence. The uncertainties in the different parameters and measurements are too large for a direct comparison of the modelled and measured daily melt rate. However, it is possible to to test the model validity by comparing the bulk melting rate, i.e. the modelled and measured cumulative loss of thickness, $\Sigma_{i=1}^{n}M_b(t_i)$.

As current velocities and iceberg temperatures are not constant during the iceberg's drift, the modelled thickness loss is fitted to the measured loss for each time step $t_i$ over a $\pm$20-day period by selecting the $V_w(t_i)$ and $T_i(t_i)$ that minimise the distance between model and observations. When no SST is available, i.e. when the iceberg is within sea ice for a short period, $T_w$ is fixed to the sea water freezing temperature.The model allows us to reproduce the thickness variations extremely well, with correlations larger than 0.999 for both B17a and C19a (see figures 6-a and 7-a) and mean differences of thickness loss of 3.1 and 0.5 m respectively and maximum differences less than 8 and 1.5 m. However, the current velocity inferred from the model,

presented in figures 6-b and 7-b, reaches very high and unrealistic values ($> 2$ m.s$^{-1}$). Compared to the altimeter geostrophic currents from AVISO, the current speed can be overestimated by more than a factor of 10.

The second model parameter $T_i$ (see figures 6-c and 7-c) varies between -20°C and -0.6°C with a $-10.9 \pm 7.1$°C mean for B17a. For C19a, it is between -9°C and 1°C with a $-10.6 \pm 5.8$ °C mean, although the model sometimes fails to converge
to realistic iceberg temperature, i.e. for $T_i < 0$°C. It happens when the measured melting is weak and SST is positive (for example from January to May 2007, figures 7-c and 5-b). The model can reproduce this inhibition by decreasing the water/ice temperature difference up to zero, resulting in an artificial increase of the iceberg temperature to positive values. For B17a, the model always converges, and the lower temperatures (-20°C) are observed during extremely rapid melting period or during the grounding period. This could reflect the decrease of ice surface temperature during rapid ablation events or an underestimation
of the melt rate.

## 4.2  Thermal turbulent exchange of Hellmer and Olbers

The second melt rate formulation is based on thermodynamics and on heat and mass conservation equations. It assumes heat balance at the iceberg-water interface and was originally formulated for estimating ice shelf melting (Hellmer and Olbers, 1989; Holland and Jenkins, 1999). The turbulent heat exchange is thus consumed by melting and the conductive heat flow
through the ice:

$$\rho_w C_{pw} \gamma_T (T_b - T_w) = \rho_i L M_b - \rho_i C_{pi} \Delta T \, M_b \tag{4}$$

Thus,

$$M_b = \frac{\rho_w C_w \gamma_T}{\rho_i} \frac{T_b - T_w}{L_H - C_{pi} \Delta T} \tag{5}$$

where $M_b$ is the melt rate (in m.s$^{-1}$), $L_H = 3.34 \cdot 10^5$ J$\cdot$kg$^{-1}$ is the fusion latent heat, $C_{pw} = 4180$ J$\cdot$kg$^{-1}\cdot$K$^{-1}$ and $C_{pi} =$
2000 J$\cdot$kg$^{-1}\cdot$K$^{-1}$ are the heat capacity of seawater and ice, respectively. $T_b = -0.0057 S_w + 0.0939 - 7.64 \cdot 10^{-4} P_w$ is the freezing temperature at the base of the iceberg, $S_w$ (in $g \cdot kg^{-1}$) and $P_w$ (in $10^4$Pa) are the salinity (here fixed at the averaged value of 35 g.kg$^{-1}$) and pressure at the bottom of the iceberg, $\Delta T = T_i - T_b$ represents the temperature gradient within the ice at the iceberg base (Jansen et al., 2007). $\gamma_T$ is the thermal turbulent velocity that can be expressed as (Kader and Yaglom, 1972)

$$\gamma_T = \frac{u^*}{2.12 \log(u^* l \nu^{-1}) + 12.5 Pr^{2/3} - 9} \tag{6}$$

where $P_r = 13.1$ is the molecular Prandtl number of sea water, $l = 1$ m the mixing length scale, $\nu = 1.83 \cdot 10^{-6}$m$^2 \cdot$s$^{-1}$ is the water viscosity, and $u^*$ the friction velocity. The latter, which is defined in terms of the shear stress at the ice-ocean boundary,

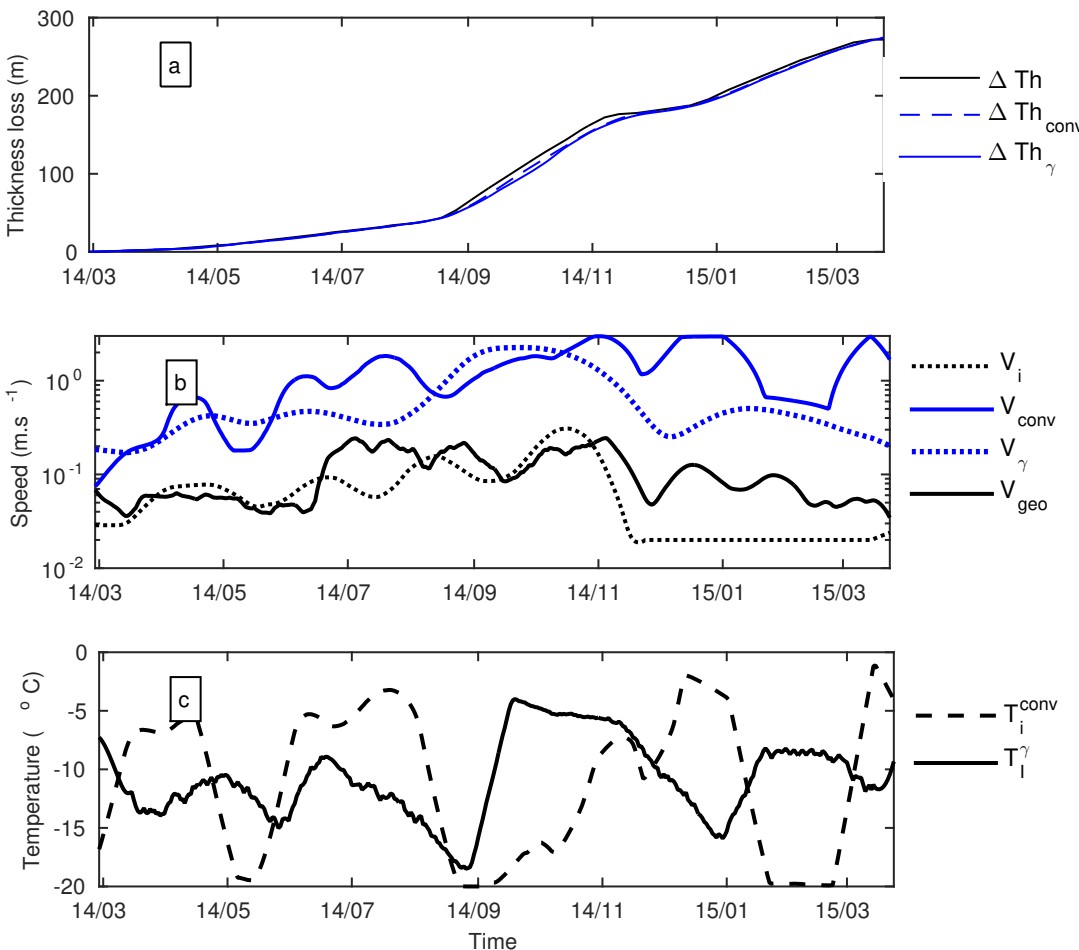

**Figure 6.** Thickness loss (in m) for B17a (a). Measured thickness loss (black line); modelled loss using forced convection (dashed blue line) and turbulent exchange (solid blue line). (b) Iceberg velocity (dotted black line). Modelled velocity using forced convection (solid blue line) and using turbulent exchange (dotted blue line). AVISO Geostrophic current velocity (solid black line). (c) Modelled iceberg temperature using forced convection (dashed line) and using thermal exchange (solid line).

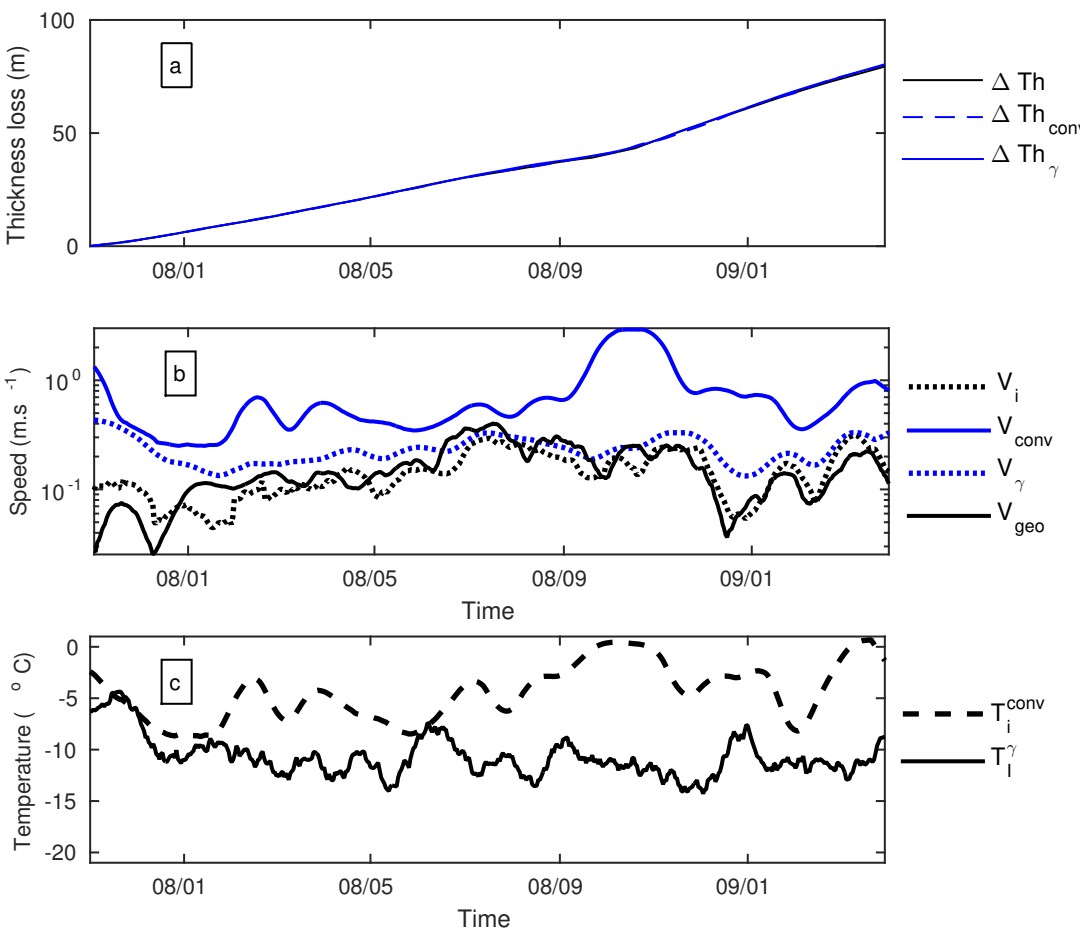

**Figure 7.** Thickness loss (in m) for C19a (a). Measured thickness loss (black line); modelled loss using forced convection (dashed blue line) and turbulent exchange (solid blue line). (b) Iceberg velocity (dotted black line). Modelled velocity using forced convection (solid blue line) and using turbulent exchange (dotted blue line). AVISO Geostrophic current velocity (solid black line). (c) Modelled iceberg temperature using forced convection (dashed line) and using thermal exchange (solid line).

depends on a dimensionless drag coefficient, or momentum exchange coefficient, $C_D = 0.0015$ and the current velocity in the boundary layer, $u \simeq V_w - V_i$, by $u^{*2} = C_D u^2$.

Jansen et al. (2007) modelled the evolution of a large iceberg (A38b) using this formulation for melting. They calibrated their model using IceSat elevation measurements and found $\gamma_T$ ranging from $0.4 \cdot 10^{-4} \mathrm{m \cdot s^{-1}}$ to $1.8 \cdot 10^{-4} \mathrm{m \cdot s^{-1}}$ close to the

$1 \cdot 10^{-4} \mathrm{m.s^{-1}}$ proposed by Holland and Jenkins (1999). Silva et al. (2006), who estimated the Southern Ocean freshwater flux by combining the NIC iceberg data base and a model of iceberg thermodynamics also based on this formulation, considered a unique and much larger $\gamma_T$ of $6 \cdot 10^{-4} \mathrm{m \cdot s^{-1}}$ .

The basal melt is thus computed using Equation 5 for $\gamma_T$ from $0.1 \cdot 10^{-5}$ to $10 \cdot 10^{-4} \mathrm{m \cdot s^{-1}}$ by $0.1 \cdot 10^{-5}$ steps and $T_i$ from -20 to 2°C by 0.1°C steps. As for forced convection, the model is fitted for each time step over a $\pm 20$ day period to estimate

$\gamma_T(t_i)$ and $T_i(t_i)$. The current speed is then estimated using Equation 6.

This model also reproduces extremely well the thickness variations with a correlation better than 0.999 for both B17a and C19a (see figures 6-b 7-a). The mean differences of thickness are 3.7 and 0.3 m for B17a and C19a respectively, and the maximum differences are 14.1 and 0.8 m. The modelled current velocity (Figures 6-b and 7-b) is always smaller than the forced convection velocity except for B17a during the three months (September to November 2014) of very rapid drift and

melting. Although it is still significantly larger than the AVISO velocities, especially for B17a, the values are more compatible with the ocean dynamics in the region (Jansen et al., 2007).

For B17a, $\gamma_T$ varies from $0.41 \cdot 10^{-4}$ to $10 \cdot 10^{-4} \mathrm{m \cdot s^{-1}}$ with a $(2.9 \pm 2.8) \cdot 10^{-4} \mathrm{m \cdot s^{-1}}$ mean. If the period of very rapid melting (September to November 2014), during which $\gamma_T$ increases up to $10 \cdot 10^{-4}$, is not considered, $\gamma_T$ varies only up to $2.5 \cdot 10^{-4} \mathrm{m \cdot s^{-1}}$ with a $(1.6 \pm 0.92) \cdot 10^{-4} \mathrm{m \cdot s^{-1}}$ mean. These values are comparable to those presented by Jansen et al.

(2007) for A38b whose size was similar to that of B17a. For C19a, $\gamma_T$ has significantly lower values ranging from $0.3 \cdot 10^{-5}$ to $1.6 \cdot 10^{-4} \mathrm{m \cdot s^{-1}}$ with $(0.34 \pm 0.37) \cdot 10^{-4} \mathrm{m \cdot s^{-1}}$ mean. These values, which correspond to the lower $\gamma_T$ found by Jansen et al. (2007), might reflect a different turbulent behaviour for very large icebergs that can more significantly modify their environment, especially the ocean circulation (Stern et al., 2016).

The mean iceberg temperature is $-10.8 \pm 5.0$°C for B17a and $-10.6 \pm 5.8$°C for C19a. It oscillates quite rapidly and certainly

more erratically than in reality.

## 4.3 Discussion

The two classical parameterisations of iceberg basal melting have been tested against observations. Both models can reproduce well the iceberg thickness variations by fitting the iceberg temperature and the current velocity. Nevertheless, the two melting strategies fail on several occasions in reproducing the observed melt rates, namely when thickness variations are important. For

instance, the forced convection approach of Weeks and Campbell (1973) requires very large current velocities and/or very high iceberg/ocean temperature difference to reproduce the measured melt rate. The large overestimation of current speed and temperature differences indicates that this model tends to underestimate the melt rate. If realistic velocities and temperatures were used, the melt rate could be underestimated by a factor of 2 to 4. This formulation is mainly a bulk parameterisation based on heat transfer over a flat plate. It was proposed in the 70's to analyse the melting of small icebergs and relies on

typical mean values of water viscosity, Prandtl number, thermal conductivity, ice density. These approximations might not be valid especially for very large tabular icebergs and can not take into account the impact of the iceberg on its environment. The velocity and temperature differences for the second formulation usually assume values that are more compatible with the ocean flow properties in the region. This parameterisation was developed for numerical models and represents the conservation of heat at the iceberg surface. It depends on both the ocean/ice and the ice surface/ice interior temperature gradients, although the ocean/ice gradient is dominant. Compared to the forced convection, for similar temperature and velocity gradients, the Hellmer and Olbers formulation leads to melt rates that are 2 to 4 times more efficient. Thus, although the current velocity can reach quite high values, this melt rate formulation is certainly better suited to reproduce the bulk melting of icebergs than forced convection. The comparison of the $M_b$ values computed using the two formulations for identical environmental parameters which shows a factor 5 difference between the forced convection and thermal turbulence for B17a ($L = 35$km) and 6-8 for C19a ($L = 150$km), confirms the underestimation of the melting by the forced convection approach.

As a consequence, our study brings out some of the limitations of the classical modelling strategies of iceberg basal melting. To make sure the second strategy is able to reproduce realistic melt rates, especially for large icebergs, we need to extend our study to more iceberg cases, namely to be able to have a broader view on the variability range of the $\gamma_T$ parameter.

## 5  Fragmentation

As said earlier, fragmentation is the least known and documented decay mechanism of icebergs. It has been suggested that swell induced vibrations in the frequency range of the iceberg bobbing on water could cause fatigue and fracture at weak spots (Wadhams et al., 1983; Goodman et al., 1980). Small initial cracks within the iceberg are likely to propagate in each oscillation until they become unstable resulting in the iceberg fracture (Goodman et al., 1980). Jansen et al. (2005) suggested from model simulations that increasing ocean temperatures along the iceberg drift and enhanced melting cause a rapid ablation of the warmer basal ice layers, while the iceberg core temperature remains relatively constant and cold. The resulting large temperature gradients at the boundaries could be important for possible fracture mechanics during the final decay of iceberg.

### 5.1  Fragmentation law

Like the calving of icebergs from glacier or ice shelves (Bassis, 2011), fragmentation is a stochastic process that makes individual events impossible to forecast. However, the probability that an iceberg will calve during a given interval of time can be described by a probability distribution. This probability distribution depends on environmental conditions that can stimulate or inhibit the fracturing mechanism (MacAyeal et al., 2006). If the environmental parameters conditioning the probability of fracture can be determined, it would thus be possible to propose at least bulk fracturing laws that could be used in numerical models. The correlation between the relative volume loss,i.e. the a-dimensional loss, $dV/V$ (which was filtered using a 20-day Gaussian window) and different environmental parameters (namely SST, current speed, difference of iceberg and current velocities, wave height, wave peak frequency and wave energy at the bobbing period) has thus been analysed in detail. The highest correlation is obtained for SST, with similar values for both icebergs, namely 0.63 for B17a and 0.64 for C19a. It is

high enough to be statistically significant and to show that SST (or the temperature difference) is certainly one of the main drivers of the fracturing process. SST is followed by the iceberg velocity which has a low correlation of 0.30 for B17a and 0.28 for C19a showing a potential second order impact. The correlation for all the other parameters, in particular for the sea state parameters, is below 0.15. Figure 8, which presents the 20 day-Gaussian filtered relative surface loss as function of SST,

iceberg velocity and wave height, confirms the strong impact of the temperature. The logarithm of the loss clearly increases almost linearly with temperature. The regression gives similar slopes of 1.06±0.04 for B17a and 0.8±0.04 for C19a. There also exists a slight increase of loss with iceberg velocity. The regression slopes are however very different for B17a (1.8±0.8) and C19a (6.3±0.8). The significant wave height has no impact on the loss.

    The cumulative sums of the relative volume loss for the two icebergs, presented in figure 9, exhibit very similar behaviour,

suggesting that a general fracturing law might exist.

    We investigate this matter step-by-step, by progressively including the dependence to environmental parameters in a simple model of bulk volume loss. Firstly, onlythe temperature difference between the ocean and the iceberg is considered in the model

$$M_{fr} = \alpha \exp(\beta(T_w - T_i)) \tag{7}$$

where $M_{fr}$ is the relative volume loss by fragmentation and $\alpha, \beta$ are model coefficients. In a first step, the daily volume loss is computed and compared to the observed loss. The model best fit presented in figure 9 (black line) gives similar results for B17a and C19a: $\alpha = 1.9 \cdot 10^{-5}$ and $2.7 \cdot 10^{-5}$, $\beta$= 1.3 and 0.91, $T_i$ = -3.4 and -3.7 $^oC$ respectively. Although the correlation between model and measurement is high (0.96 and 0.98 respectively), the model does not reproduce very well the final decay of the iceberg.

A possible second order contribution of the iceberg velocity is thus taken into account by introducing a second correction term in the model in the form:

$$M_{fr} = \alpha \exp(\beta(T_w - T_i))(1 + \exp(\gamma V_i)) \tag{8}$$

The model is first fitted by setting the $\beta$ coefficient to the value found using the simple model. The best fit of the model is presented as a blue line in figure 9. The fitting parameters have quite similar values for the two icebergs, $\alpha = 5 \cdot 10^{-6}$for both,

$\gamma$= 5.3 and 6.2 and $T_i$ = -3.3 and -4 $^oC$ respectively. The inclusion of velocity clearly improves the modelling of the final decay and increases the correlation to more than 0.99.

    The possibility of a general law has been further investigated by testing the model with a common $\beta$ of 1 for both icebergs. The best fit is presented as green lines. The best fit is only slightly degraded (correlation about 0.992). The $\gamma$ and $T_i$ fitting parameters slightly vary and are of the same order of magnitude for the two icebergs. Only the $\alpha$ parameter strongly differs for

B17a ($3 \cdot 10^{-5}$) and C19a ($5 \cdot 10^{-6}$). This can result from the fact that the variability of iceberg temperature is not taken into account. Indeed, a change of $T_i$ of $\Delta T$ introduces a change of $\alpha$ of $\exp(-\beta \Delta T)$.

| Iceberg | B17a | | | | C19 | | | |
|---|---|---|---|---|---|---|---|---|
| Model /Parameters | $\alpha$ | $\beta$ | $T_i$ | $\gamma$ | $\alpha$ | $\beta$ | $T_i$ | $\gamma$ |
| 1- $\alpha\exp(\beta(T_w - T_i))$ | $\mathbf{1.9 \cdot 10^{-5}}$ | **1.3** | **-3** | | $\mathbf{2.7 \cdot 10^{-5}}$ | **0.91** | **-3.7** | |
| 2- $\alpha\exp(\boldsymbol{\beta}(T_w - T_i))(1 + \exp(\boldsymbol{\gamma}V_i))$ | $\mathbf{5.0 \cdot 10^{-6}}$ | 1.3 | **–3.3** | **5.3** | $\mathbf{5.0 \cdot 10^{-6}}$ | 0.91 | **-4** | **6.2** |
| 3- $\alpha\exp(\beta(T_w - T_i))(1 + \exp(\gamma V_i))$ | $\mathbf{3.0 \cdot 10^{-6}}$ | 1 | **-3** | **6** | $\mathbf{5. \cdot 10^{-6}}$ | 1 | **-3.2** | **7.2** |
| 4- $\alpha\exp(\beta(T_w - T_i))(1 + \exp(\gamma V_i))$ | $1.0 \cdot 10^{-6}$ | 1 | Piecewise | 6.5 | $1.0 \cdot 10^{-6}$ | 1 | Piecewise | 6.5 |

**Table 1.** Fragmentation models and parameters. The bold characters represent fitted parameters while the regular characters represent the fixed values.

A final model is tested in the same way as the melting law. The $\alpha$, $\beta$ and $\gamma$ parameters are fixed at $1 \cdot 10^{-6}$, 1 and 6.5 respectively, and the model is fitted at each time step over a $\pm 20$ day period to determine the best fit $T_i$. The model fit the data with correlation higher than 0.998. The iceberg temperature varies by less than $2^oC$ and has a mean of $-3.7 \pm 0.6^oC$ for B17a and $-2.9 \pm 0.6^oC$ for C19a (see figure 10). Table 1 summarises the different models and fitted parameters for the two icebergs. Other model formulations including wave height, iceberg speed and wave energy at the bobbing period were tested but brought no improvement.

## 5.2 Transfer of volume and distribution of sizes of fragments

The fragmentation of both icebergs generates large plumes of smaller icebergs that drift on their own path and disperse the ice over large regions of the ocean. The knowledge of the size distribution of the calved pieces is as important as the fragmentation law for modelling purposes as the fragment size will condition their drift and melting and ultimately the freshwater flux. The fragment size distribution is analysed using both the ALTIBERG small icebergs database and the analysis of three clear MODIS images that present large plumes of pieces calved from C19a and B17a. Figures 11-a and c present the small icebergs detected by altimeters in the vicinity (same day and 400 km in space) of B17a and C19a. To restrict as much as possible a potential influence of icebergs not calved from the one considered, the analysis of the iceberg size is restricted to the period when C19a drifted thousands of kilometres away from any large iceberg. During this period more than 2400 icebergs were detected. The corresponding size distribution is presented in figure 13.

The small iceberg detection algorithm used to analyse the MODIS images is similar to those used to estimate the large iceberg area. Firstly, the cloudy pixels are eliminated by using the difference between channel 1 and 2 radiances. The image is then binarised using a radiance threshold. A shape analysis is then applied to the binary images to detect and characterise the icebergs. The results are then manually validated. Figure 12 presents an example of such a detection for C19a. The full resolution images are available in the Supplementary Information (Figures S1 to S4). The analysis detected 1057, 817, 1228 and 337 icebergs for the four images respectively. The size distributions for the four images and for the overall mean are given also in figure 13. The six distributions are remarkably similar between 0.1 and 5 km$^2$. The tail of the distributions (i.e. for area larger than 7 km$^2$) is not statistically significant because too few icebergs larger than 5-6 km$^2$ were detected.

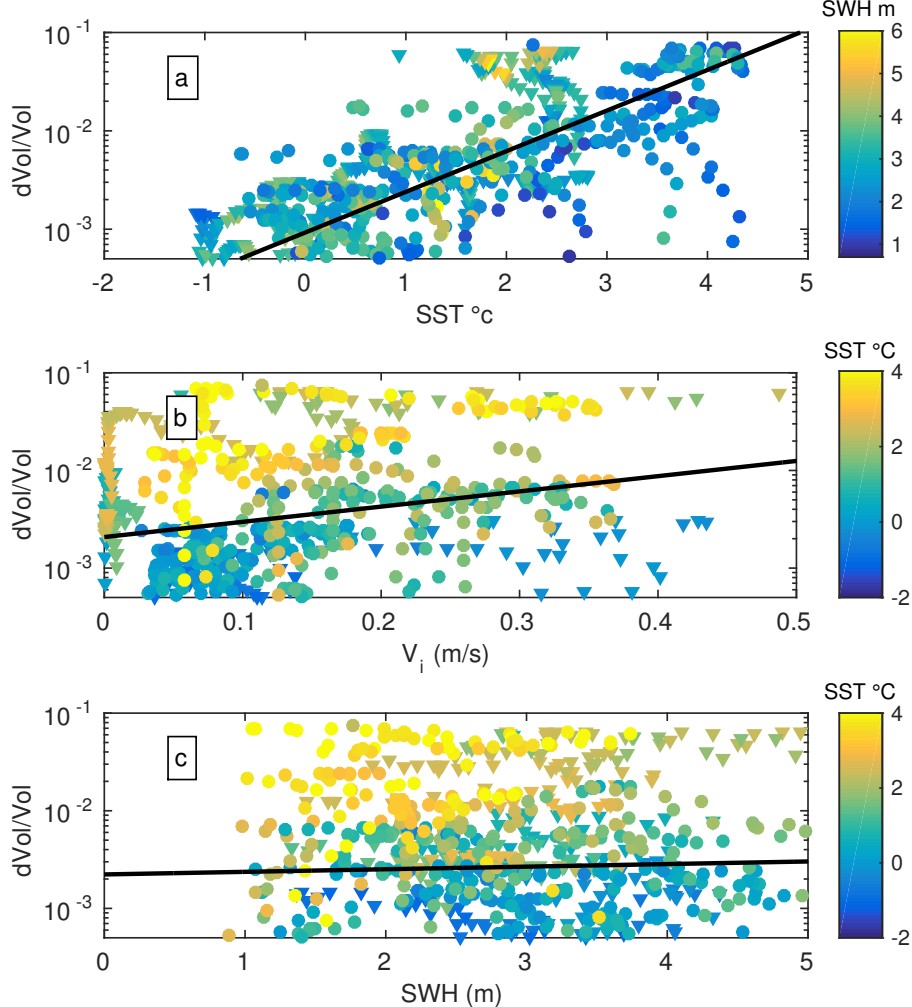

**Figure 8.** (a) Relative volume loss $dV/V$ as a function of SST. The colour represents the significant wave height in m. (b) $dV/V$ as a function of the iceberg velocity. The colour represents the SST in °C. (c) $dV/V$ as a function of significant wave height. The circles correspond to C19a and the triangles to B17a. The red lines represent the regression lines. The ordinate scale is logarithmic.

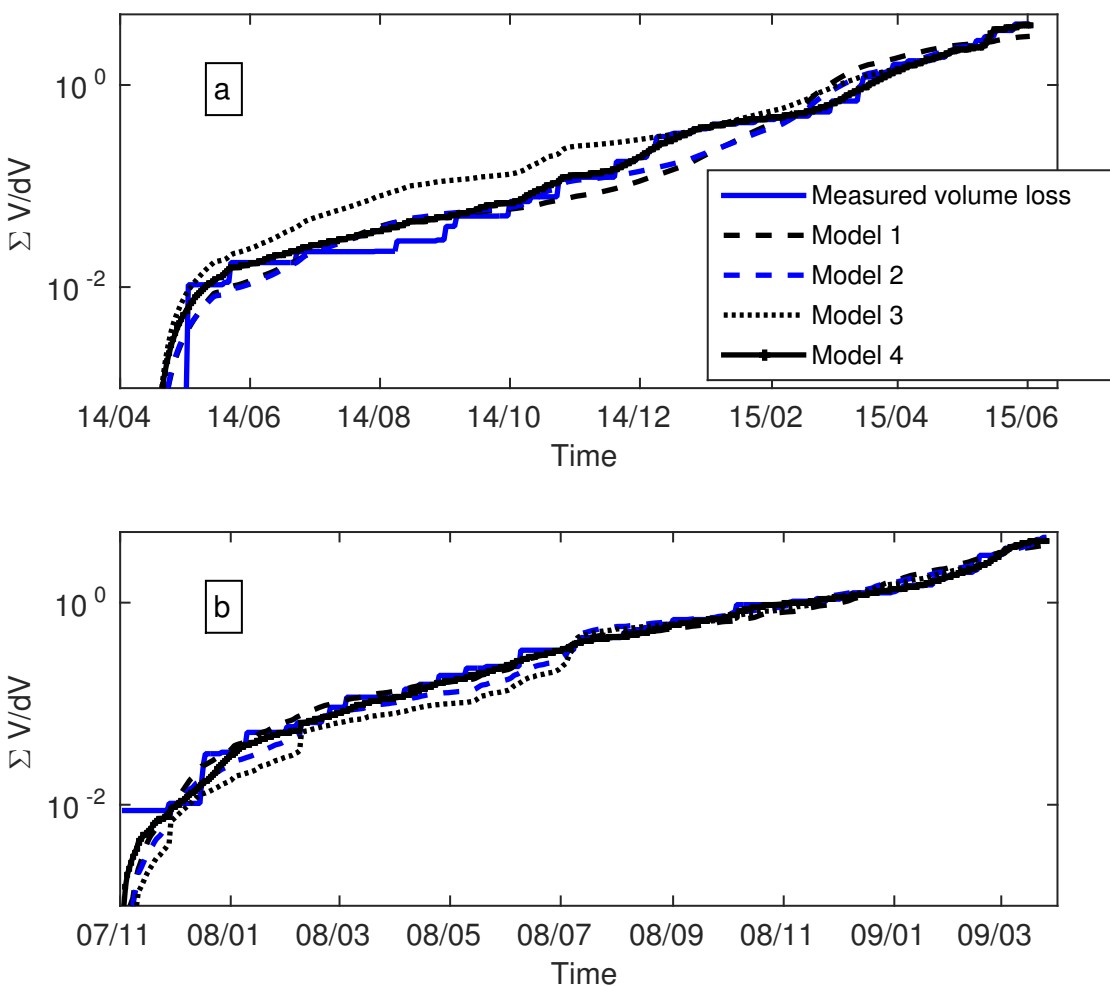

**Figure 9.** (a) Cumulative relative volume loss, $\sum dV/V$, measured (solid blue line), modelled depending on temperature difference only (dashed black line), on temperature difference and iceberg velocity (dashed blue line), on temperature difference and iceberg velocity with $\beta = 1$ (dotted black line), full model fitted piece-wise (solid black line). (a) B17a, (b) C19a. The ordinate scale is logarithmic.

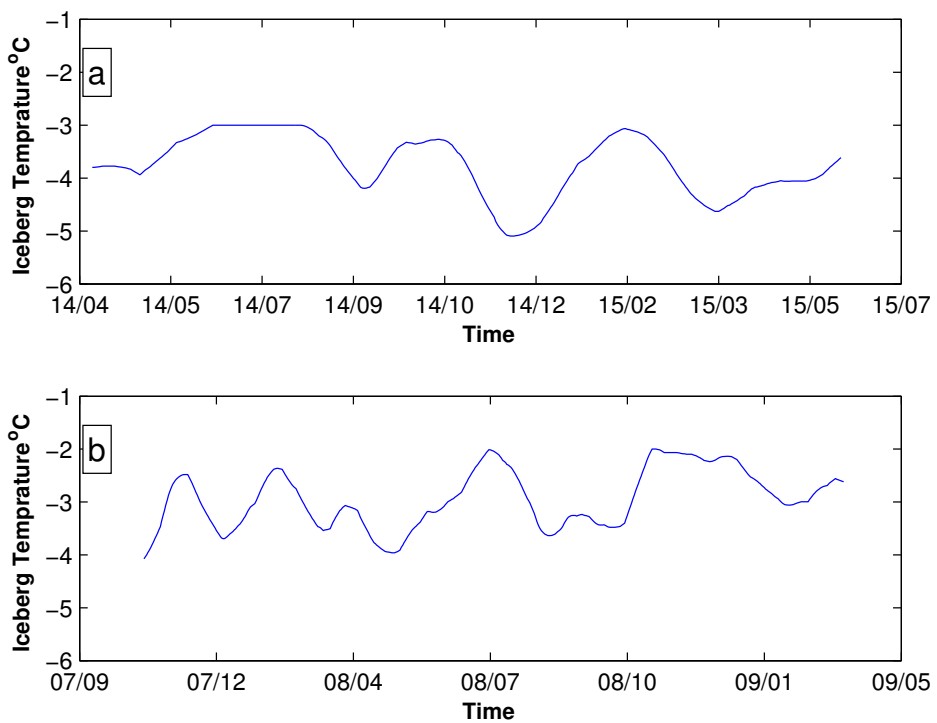

**Figure 10.** Fitted iceberg temperature for B17a (a) and C19a (b).

The slopes of the distributions have thus been estimated by linear regression for areas between 0.1 and 5km$^2$. The values for the four images are -1.49±0.13, 1.63±0.15, -1.41±0.15, -1.44±0.24 respectively and 1.53±0.12 for the overall mean distribution. The slope of the ALTIBERG iceberg distribution is -1.52±0.07. These values are all close to the -3/2 slope previously presented by Tournadre et al. (2016) for icebergs from 0.1 to 10,000 km$^2$. A -3/2 slope has been shown both
5   experimentally and theoretically to be representative of brittle fragmentation (Astrom, 2006; Spahn et al., 2014).

This size distribution represents a statistical view of the fragmentation process over a period of time that can correspond to several days or weeks. Indeed, it is impossible to determine from satellite image analysis or altimeter detection the exact calving time of each fragment, and it is thus impossible to estimate the exact distribution of the calved pieces at their time of calving. In the same way as fragmentation is characterised by a probability distribution, the size of the fragment will also
10   be characterised by a probability distribution. The size distribution represents the integration over a period of time of this probability distribution. It can be used to model the transfer of volume calved from the large iceberg to small pieces.

The transfer of volume from the large icebergs to smaller pieces can also be estimated using the small iceberg area data from the ALTIBERG database. The sum of the areas of the detected fragments is presented in figure 11-b and d as well as the large iceberg surface loss by fragmentation. The difference between the two curves can result from : 1) an underestimation of

the number of small icebergs, 2) the total area of pieces larger than ~8 km$^2$ not detected by altimeters. While 1 is difficult to estimate, 2 can be computed, assuming that the piece distribution follows a power law. Annex A2 presents the details of the computation. For both icebergs, as long as the surface loss is limited, the number of calved pieces is small and the probability for a fragment to be too large to be detected by altimeter is also small. The total surface of the detected small icebergs represents thus almost all the parent iceberg surface loss. As the degradation increases so does the surface loss. The number of calved pieces as well as the probability of larger pieces calving become significantly larger resulting in a larger proportion of the surface loss due to pieces larger than 8 km$^2$ (thus not detected). The overall proportion of the surface loss due to small icebergs is about 50% in good agreement with the power law model of Annex A2.

ashed

# 6   Conclusions

The evolution of the dimensions and shape of two large Antarctic icebergs was estimated by analysing MODIS visible images and altimeter measurements. These two giant icebergs, named B17a and C19a, were worthy of interest because they have drifted in the open ocean for more than a year, are relatively remote from other big icebergs, and were frequently sampled by our sensors (altimeters and MODIS). Furthermore, the two of them exhibited very different features, in terms of size and shape as well as in their drift characteristics. We thus expect their joint study to be an opportunity to obtain a more comprehensive insight into the two main processes involved in the decay of icebergs, melting and fragmentation.

Basal melting is the main cause of an iceberg's thickness decrease. The two main formulations employed to represent the melting of iceberg in numerical models have been confronted to the evolution of the iceberg thickness. The two melting models, which differ in their formulation depend primarily on the same two quantities: the iceberg/water differential velocity and their temperature difference. The classical bulk parameterisation of the forced convection is shown to strongly underestimate the melt rate, while the forced convection approach, based on the conservation of heat, appears better suited to reproduce the iceberg thickness variations.

The main decay process of icebergs, fragmentation, involves complex mechanisms and is still poorly documented. Due to the stochastic nature of fragmentation, an individual calving event cannot be forecast. Yet, fragmentation can still be studied in terms of a probability distribution of a calving. We carried out a sensitivity study to identify which environmental parameters that likely favour fracturing. We thus analysed the correlation between the relative volume loss of an iceberg and some environmental parameters. The highest correlations are found firstly for the ocean temperature and secondly for the iceberg velocity, for both B17a and C19a. All other parameters (namely the waves-related quantities) show no significant link with the volume loss. We then formulated two bulk volume loss models: firstly one that depends only on ocean temperature, and secondly one that takes into account the influence of both identified key parameters. The two formulations are fitted to our relative volume loss measurements, and the best fitting parameters are estimated. Using iceberg velocity along with ocean temperature clearly better reproduces the volume loss variations, especially the quicker ones seen near the final decays of both bergs. Moreover, if

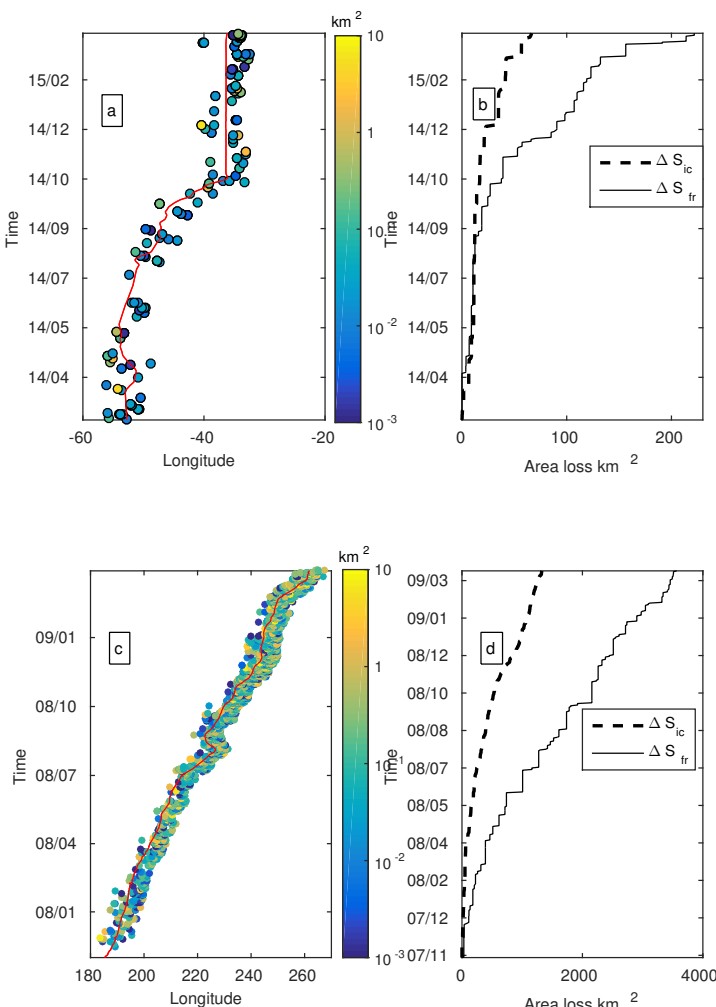

**Figure 11.** Time/longitude trajectory of B17a (a) and C19a (c) (red line) and coincident small icebergs detected in its vicinity. The colour represents the area of the iceberg in log scale. Surface loss by breaking (black lines) and surface of the detected small icebergs (dashed line) for B17a (b) and C19a (d).

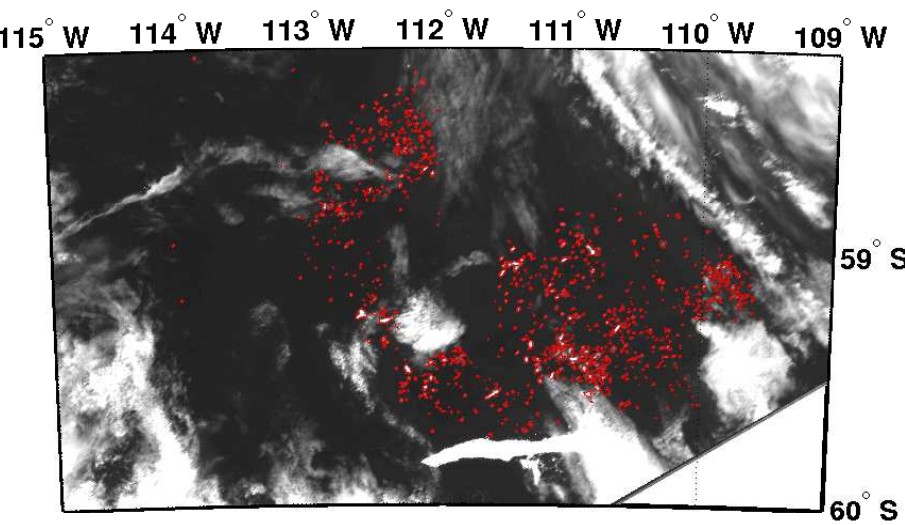

**Figure 12.** Example of fragment detection using a MODIS image (C19a 02/05/2009). The contour of the detected icebergs are represented in red lines.

the variability of the iceberg temperature is taken into account, the model coefficients are in this case quite similar for the two icebergs.

Finally, we have estimated the size distribution of the fragments calved from B17a and C19a, using MODIS images and altimetry data. For both icebergs and both methods, the slope of the distribution is close to -3/2, consistent with our previous

5  altimetry-based global study and typical of brittle fragmentation processes.

While giant icebergs are not included in the current generation of iceberg models, they transport most of the ice volume in the Southern Ocean. Furthermore, the impact of icebergs on the ocean in global circulation models strongly depends on their size distribution (Stern et al., 2016). As a consequence, it is believed that the current modelling strategies suffer from a "small iceberg bias". To include large icebergs in models would require us to ascertain that the previous modelling strategies are still

10  valid for large icebergs. We also ought to gain more knowledge on how these bigger bergs constrain a size transfer to produce medium to small pieces via fragmentation. Eventually, these smaller pieces are those that account for the effective fresh water flux in the ocean. Our study showed that a classical modelling strategy is able to reproduce the basal melting of large icebergs, provided that relevant parameters are chosen. It has also demonstrated that a simple bulk model with appropriate environmental parameters can be used to account for the effect of the fragmentation of large icebergs, and highlighted the consequent size

15  distribution of the pieces. These results could prove valuable for including a more realistic representation of large icebergs in

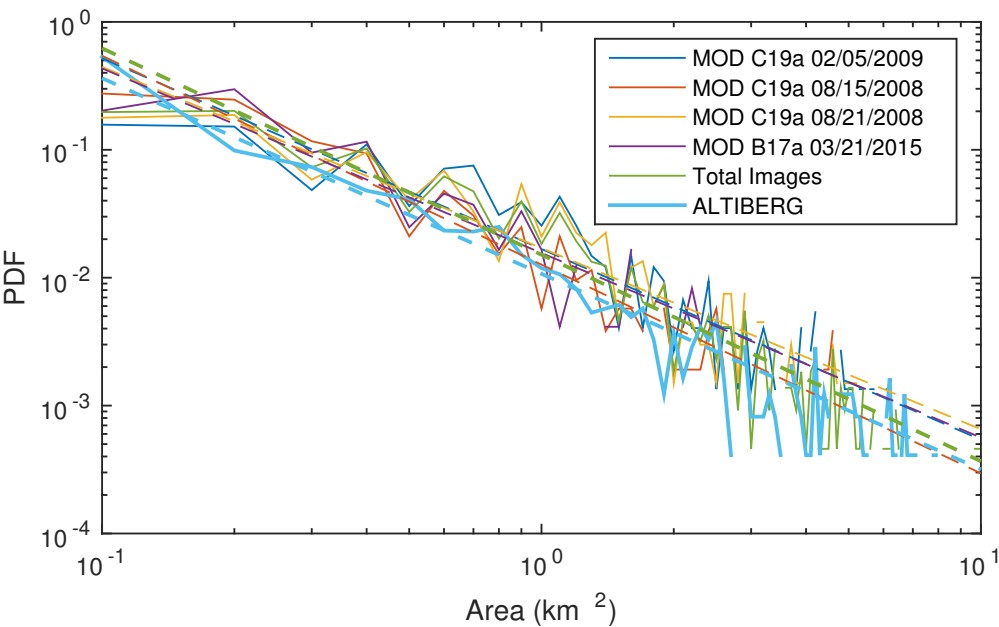

**Figure 13.** Probability density function of the fragment size detected on MODIS images (dark blue line C19a 02/05/2009, orange line C19a 08/15/2008, yellow line 08/21/200, violet line B17a 03/02/2015, green line all images), and detected by altimeter in the vicinity of C19a ( turquoise line). The dashed straight lines represent the power law fit to the data.

models. Our analyses could be extended to the cases of more large icebergs, namely to validate our bulk modelling approaches on a more global scale.

*Acknowledgements.* The MODIS images were provided by NASA through the LAADS DAAC ( http://ladsweb.nascom.nasa.gov/). The altimeter data were provided by the french Centre National d'Etude Spatiale (CNES), the European Space Agency (ESA), EUMETSAT, the US National Aeronautics and Space Administration (NASA), the US National Oceanic and Atmospheric Administration and the Chinese National Ocean Satellite Application Center (NSOAS). The geostrophic current were provided by the AVISO center (https://www.aviso.altimetry.fr). The large iceberg tracks were provided by the Brigham Young University Center for Remote Sensing (http://www.scp.byu.edu/data/iceberg/database1.html) and the National Ice Center (http://www.natice.noaa.gov/). The ALTIBERG data sets are available at ftp://ftp.ifremer.fr/ifremer/cersat/projects/altiberg. The study was partially founded by CNES through the TOSCA program.

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

**A2   Power law and total area distribution**

The fragment size probability follows a power law with a -3/2 slope for sizes between $s_1$ and $s_2$ thus

$$P(s) = \alpha_0 s^{-3/2} \tag{A2}$$

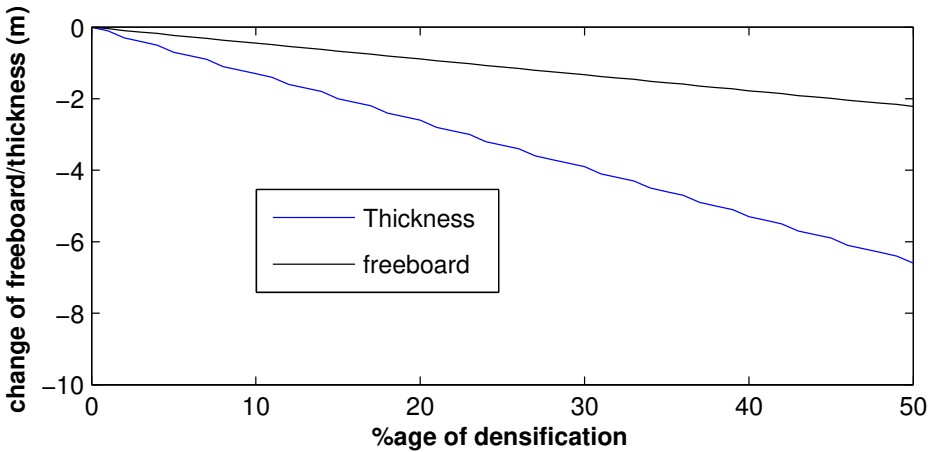

**Figure A1.** Variation of thickness (blue line) and freeboard (black line) as a function of the percentage of firn densification for a 450 m thick iceberg

where $\alpha_0 = \sqrt{s_0 s_1}/(2(\sqrt{s_1} - \sqrt{s_0}))$.

If $N_0$ is the number of calved icebergs of sizes between $s_3$ and $s_4$, then the distribution of the number $N$ is

$$N(s) = N_0 \alpha_0 s^{-3/2}$$

The maximum iceberg size $s_{lim}$, i.e. the class for which $N(s_{lim}) = 1$, is $s_{lim} = (N_0 \alpha_0)^{2/3}$. The proportion of the total surface represented by the icebergs of sizes between $s_3$ and $s_4$ is thus

$$R(N_0) = \frac{\int_{s_3}^{s_4} N_0 \alpha_0 s\, s^{3/2} ds}{\int_{s_1}^{s_{lim}} N_0 \alpha_0 s\, s^{3/2} ds} = \frac{\sqrt{s_4} - \sqrt{s_3}}{\sqrt{(N_0 \alpha_0)^{2/3}} - \sqrt{s_1}} \tag{A3}$$

Figure A2 presents $R$ for $s_4$ from 4 to 9 km$^2$, $s_1 = 0.01$km$^2$, i.e. the smallest iceberg detectable using MODIS, $s_3 = 0.1$km$^2$, i.e. the detection limit of altimeter, $s_2$ has been set to 40 km$^2$, size of the largest piece detected on the MODIS images. If a thousand fragments have been created, icebergs smaller than 6 km$^2$ represent only 60% of the total surface, the icebergs smaller than 8 km$^2$ account for 70%. For 2000 fragments, the proportion drops to 50 and 55% respectively.

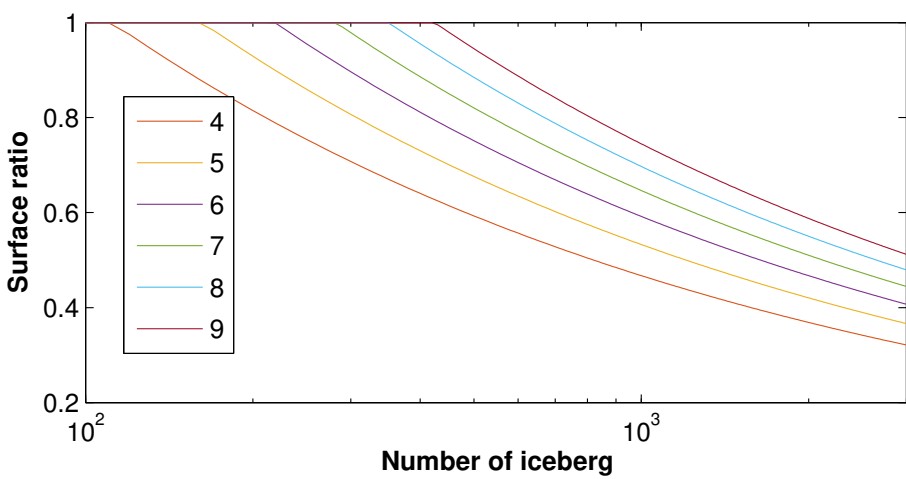

**Figure A2.** Proportion of the total surface represented by icebergs of area between 0.1 and 4 to 9 km$^2$ as a function of the total number of icebergs.