# Peer review of "Melting and fragmentation laws from the evolution of two large Southern Ocean icebergs estimated from satellite data"

_The Cryosphere, 2017_

## Referee Comment (RC1) · Anonymous Referee #1 · 14 Dec 2017

GENERAL COMMENTS:

This manuscript presents an interesting analysis of the decay of two large icebergs, as tracked from satellite imagery and from altimetry. The ice bergs both decrease in size over time. The data products together show thickness and horizontal extent of the ice bergs, so the investigators are able to distinguish mass loss due to melt (change in thickness) vs fragmentation (changes in surface area). Results show that fragmentation is the major source of mass loss for large ice bergs. The study then assesses the details of melting and fragmentation. The authors compare two melt models, one associated with thermodynamics and a second based on thermodynamics, and find that the turbulent thermodynamic model better represents the observations. They then examine the statistics of fragmentation by looking at pdfs of ice berg sizes. I particularly

appreciate the assessment of the results in the context of both melting and fragmentation theories.

The analysis is thorough and the results are likely to attract broad interest. However, it will require more careful editing prior to publication. My detailed comments follow:

I have reviewed the list of 15 questions in the instructions for reviewers, and of these questions, only a handful raise concerns, as follows:

(5) Are the results sufficient to support the interpretations and conclusions? For the most part results are sufficient. It would be preferable to include formal uncertainty estimates in the figures, where possible.

(11) Is the language fluent and precise? As noted below, the language could be improved.

(12) Are mathematical formulae, symbols, abbreviations, and units correctly defined and used? As noted below, mathematical symbols have suffered from a font substitution problem, which should be corrected if possible in future pdf versions.

............ SPECIFIC COMMENTS:

(1) For this assessment of time series of iceberg properties, assessment of uncertainty are important and should be included. In particular, some of the quantities in Figures 4 and 5 should be plotted with uncertainties. Others might be OK with uncertainties indicated in the figure caption.

(2) The summary raises some interesting points about possible overestimates in previous studies of freshwater flux due to icebergs. But I haven't found a clear estimate of freshwater flux in this study. Is it possible to provide a concrete number?

(3) Mathematical notation is not type-set with embedded fonts. My print out has "square root" symbols in place of "less than or equal" and "greater than or equal" signs for the inequalities on line 6 of p. 2. Likewise, vectors in equation (3 have lost their arrowheads
in my printout. For a journal that people read from downloaded pdfs, this is especially challenging. I admit that the cryosphere models are outside my domain of expertise, and I have trusted the authors on these and have not attempted to work through the background literature to evaluate their appropriateness.

DETAILED COMENTS:

(4) References are not cited correctly in many places, and a number of references that should be in the main text have been embedded in parentheses. This needs to be proofread carefully for citation style.

(5) The writing, while generally OK, includes passages that are difficult to understand. I cannot possibly identify everything, and I encourage the authors to locate a native English speaker who can help them proofread in detail.

(a) Here is my suggested rewriting of the abstract:

The evolution of the thickness and area of two large Southern Ocean icebergs that have drifted in open water for more than a year is estimated through the combined analysis of altimeter data and visible satellite images. The observed thickness evolution is compared with iceberg melting predictions from two commonly used melting formulations, allowing us to test their validity of large icebergs. The first formulation, based on a fluid dynamics approach, tends to underestimate basal melt rates, while the second formulation, which considers the thermodynamic budget, appears more consistent with observations. Fragmentation leads melting as the major process responsible for the decay of large icebergs. Despite its importance, fragmentation remains poorly documented. The correlation between the observed volume loss of our two icebergs and environmental parameters highlights factors most likely to promote fragmentation. Using this information, a bulk model of fragmentation is established that depends on ocean temperature and iceberg velocity. The model is effective at reproducing observed volume variations. The size distribution of the calved pieces is estimated using both altimeter data and visible images and is found to be consistent

with previous results and typical of brittle fragmentation processes. These results are valuable in accounting for the freshwater flux constrained by large icebergs in models.

(b) p. 1, lines 17 and 18. What is meant by "buffer"? What is meant by "diffuse"? Should it be "diffusive"?

(c) Language is in places too informal, with the use of contractions. For example, line 9 of p. 2: "can't" should be "cannot" in formal writing.

(d) p. 2m lines 23-25. Change wording and punctuation to clarify:

" identified three styles of calving during the drift: "rift calving", which corresponds to the calving of large daughter icebergs by fracturing along preexisting flaws; "edge wasting", the calving of numerous small edge-parallel, sliver-shaped small icebergs; and "rapid disintegration", which is characterised by the rapid calving of numerous icebergs."

(e) p. 2, line 34. "allow to" –> "allow us to"; also p. 9, line 20. (It's a transitive verb.) Likewise for p. 4, line 19. "enables" –> "enables us"

(f) "ones" should not be used as a substitute noun in a comparison. For example, on p. 4, just before heading 2.2, you can say "For example, B17a was sampled by 152 altimeter passes during its drift and C19a by 258 passes." The word "ones" is unclear and non-standard English. Another example on p. 9, line 17: "measured one"–>"measured loss".

(g) p. 9, line 3. Does it take several years for the iceberg surface temperature to depend on the ablation rate? Or should this say, "Icebergs can sometimes float for several years. After initial adjustment, the iceberg surface temperature depends on the ablation rate."

(h) p. 1, line 16. Rewrite to say, "However, their melting accounts for less than 20% of their mass loss, and the majority of ice loss (80%) is achieved through breaking into smaller icebergs (Tournadre et al., 2016)."

(i) p. 1, line 21. The word "between" doesn't seem clear hear. Perhaps the authors mean, "differ in their basal ice-shelf and iceberg melting" or "achieve different relative balances of basal ice-shelf and iceberg melting"?

(j) p. 2, line 13. "law" –> "laws"

(k) p. 3, line 10. Capitalize "Southern Ocean"

(l) p. 3, line 11. Not clear if this is a singular or plural noun. To clarify perhaps, "area, size, and shape have been estimated".

(m) p. 3, line 14. By saying "that drifted", this implies that there are a number of other icebergs that drifted for more than two years in the South Pacific. I think the meaning would be clearer if "that" were replaced with "and".

(n) p. 3, line 15. "a relatively small 200 kmˆ2 one drifting" –>"relatively small (200 kmˆ2) and drifted"

(o) p. 3, line 15. To clarify the distinction between the plums and the big icebergs, say "both large icebergs".

(p) p. 3, line 5(2nd case). "confronted to" –> "confronted with".

(q) p. 4, line 21. "small icebergs location" –> "small iceberg locations". (The word "iceberg" is used as an adjective, and nouns used as adjectives are nearly always singular.)

(r) p. 4, line 8. "ones" –> "passes"

(s) p. 4, line 20. "constrains" –> "constraints"

(t) p. 4, line 18. "as iceberg" –> "as an iceberg"

(u) p. 5, line 17. Try "For each image with good cloud clover and light conditions ...."

(v) p. 5, line 28. "proxy of" –> "proxy for"

(w) p. 9, line 4. Try "can theoretically warm up to ...."

(x) p. 9, line 9. "shows" –> "show" .... "ones" –> "velocities"

(y) p. 9, line 10. "thus considered as" –> "treated as"

(z) p. 9, line 17. "one" –> "loss"

(aa) p. 9, line 25. Try "The second model parameter Ti (see Figures 6-c and 7-c) varies between -20âŮ̧eC and -0.6âŮ̧eC for B17a, with a $-10.9{\pm}7.1$âŮ̧eC mean for B17a. For C19a, it is between -9âŮ̧eC and 1âŮ̧eC, with a $-10.6{\pm}5.8$âŮ̧eC, although the model sometimes fails to converge to realistic iceberg temperature, i.e. for Ti<0âŮ̧eC. This occurs ...."

(bb) p. 9, line 34. "fail" –> "fails"

(cc) p. 11, line 29 Try "calving of icebergs from glaciers or ice shelves"

(dd) p. 12, line 13. "exist" –> "exists"

(ee) p. 12, line 17. "We investigate this matter by progressively including the dependence on environmental parameters ...."

(ff) p. 12, line 23. "ones" –> "loss"

(gg) p. 13, line 13. Clearer wording perhaps: "tested but brought no improvement".

(hh) p. 13, line 35. Missing superscript for km^2

(ii) p. 13, line 17. "fragments" –> "fragment"

(jj) p. 14, line 22. "in open ocean" –> "in the open ocean"

(kk) p. 14, line 22 "relatively" –> "are relatively"

(ll) p. 14, line 24. "get" –> "obtain" (The word "get" sounds too colloquial for formal writing.)

(mm) p. 14, line 29 Try "the first is more dynamically based, and the second results from a thermodynamic balance"

(nn) p. 15, line 7. "chose to carry" –> "carried", "find out which" –> "identify the", "parameters are more likely to" –> "parameters that likely favour"

(oo) p. 15, line 23 onward. I'm not sure what is meant by this discussion. Change to "small iceberg bias". Does the "them" in "To include them" refer to small or large icebergs? The context suggests large icebergs, but the wording implies small icebergs. The word "still" should be removed.

(pp) p. 15, line 5-7. I would remove the "On the one hand" /"On the other hand" structure. It's a bit informal and doesn't clarify the meaning. The second sentence can begin "It also has demonstrated ...."

---

## Referee Comment (RC2) · Anonymous Referee #2 · 20 Dec 2017

This paper presents an assessment of two different melting model approaches for icebergs during their drift and introduces an empirical fragmentation law developed from observations for the fracturing processes during iceberg drift. In principle, this is an interesting story with potential for improving models of freshwater input into the Southern Ocean by iceberg melting. However, the manuscript needs some work before publishing. A main problem is a lack of structure, which makes the line of thought hard to follow for the reader.

The manuscript contains a lot of different types of observational data, models, model results, so that it might have been better to divide the story into two manuscripts. The introduction is rather long and detailed, but at the same time is lacking a clear line of thought. It should be more concise, and more importantly make the contribution or the

potential of the presented approach to larger research goals more clear. The rest of the manuscript does not follow the usual methods/ results/ discussions structure. After the introduction a "data" section follows, after which already the results from the iceberg observations are presented. Then one melt model is introduced, and the results presented, before the second model approach is described. This is not good for the reading flow. My suggestion for a better structure would be to clearly divide the paper in two parts, 1. Melting, 2. Fragmentation, and follow a classic methods/ result / discussion structure in each of the separate parts. The introduction and an additional joint discussion then should make it clear how these parts belong together. In the first part you could have a "methods" chapter where the remote sensing data and their analysis is described, and the two melt approaches as well as the explanation that you are assessing and comparing their performance. Then present the melt results from observations, and the model results. Followed by a discussion of all three results. In a second part the fragmentation model could be explained. In this way the paper could be made more concise and clearer. The summary is too long and repetitive, and contains parts which should be mentioned before in a discussion. In my view it would be necessary to thoroughly rework the structure of the paper in order to communicate the actual value of the study.

Language: There are problems with punctuation, grammar and expressions in some places, which need a revision and maybe a read-through by a native speaker.

Figures: The labels on the axes should appear as the same font size, and should not overlap as in figure 4. I would suggest including a table with the fitting parameters for the fragmentation law, instead of printing the equations into the figure. Where possible I would place the panel labels outside of the main plot area and without a frame.

Specific comments: Title: the title is not an adequate description of the content, as previously established melt models are being assessed, and a new empirical law for fragmentation is presented. Southern Ocean is a name and should be written with capital letters.

Page 1, line16: "melting accounts for less than 20 % of their mass loss" This is only true for the final stages of decay, as most large tabular icebergs keep their shape quite well during drift. This expression is also a bit misleading, as in the end all mass is lost due to melting, as also the smaller icebergs do melt.

Page 3, line 13: please insert the web-address of the BYU data base as a reference.

Page 5, line 15: insert "Here," in front of the second sentence to make it clear that now you are talking about your study and no longer about the BYU data.

Page 5, line 37: "Due to lack of a better alternative…SST is used": I understand that the ocean temperature at the base of the iceberg, i.e. in about 300m depth, is not easy to obtain, but it would be necessary to at least discuss this as an error source and get some data from models to estimate the difference possible between the t1emperature at the surface and at depth.

Page 6, line 9: I think SI units are standard for TC.

Page 7, first paragraph: Something that is completely missing here is the influence of firn compaction or changes in density along drift. This can have a substantial effect on the freeboard of an iceberg, while no mass is lost. This should be explained and an error should be estimated for this. There is also no information about which mean density has been assumed and why. The units in this and the following paragraphs are not displayed correctly in the pdf.

Page 10, line 16: "melt rate"

Page 10, last paragraph: here methods, results and discussion are all mixed up in one single paragraph.

Page 11, line 18: "melt rate"

Page 12, first paragraph: here also all in one paragraph: first discussion and interpretation of results ("highest correlation is obtained for…"), and only after this a reference

to the figure where the data is shown. This is confusing to read. First describe the results, then interpret and discuss them

Page 12, line 17-19: This paragraph is unclear, are there words missing? ("volume loss depending"?)

Page 13, first paragraph: As the model is derived by fitting the observations, it should not be a surprise that there is a good correlation. It would have been interesting to discuss the meaning of the fit parameters, and why they are different for the two icebergs. In my view a useful empirical model should be able to reproduce the fragmentation of different icebergs with the same parameters.

---

## Author Comment (AC1) · 13 Feb 2018

[british,english]article

**Reply to referee's comments**

February 13, 2018

The authors wish to thank referee 1 who spent time to correct many grammar and spelling mistakes and to provide us with many useful comments.

**Anonymous Referee #1**

Received and published: 14 December 2017

**GENERAL COMMENTS:**

This manuscript presents an interesting analysis of the decay of two large icebergs, as tracked from satellite imagery and from altimetry. The ice bergs both decrease in size over time. The data products together show thickness and horizontal extent of the ice bergs, so the investigators are able to distinguish mass loss due to melt (change in thickness) vs fragmentation (changes in surface area). Results show that fragmentation is the major source of mass loss for large ice bergs. The study then assesses the details of melting and fragmentation. The authors compare two melt models, one associated with thermodynamics and a second based on thermodynamics, and find that the turbulent thermodynamic model better represents the observations. They then
examine the statistics of fragmentation by looking at pdfs of ice berg sizes. I particularly appreciate the assessment of the results in the context of both melting and fragmentation theories.

The analysis is thorough and the results are likely to attract broad interest. However, it will require more careful editing prior to publication. My detailed comments follow:

I have reviewed the list of 15 questions in the instructions for reviewers, and of these questions, only a handful raise concerns, as follows:

(5) Are the results sufficient to support the interpretations and conclusions? For the most part results are sufficient. It would be preferable to include formal uncertainty estimates in the figures, where possible.

(11) Is the language fluent and precise? As noted below, the language could be improved.

(12) Are mathematical formulae, symbols, abbreviations, and units correctly defined and used? As noted below, mathematical symbols have suffered from a font substitution problem, which should be corrected if possible in future pdf versions.

We checked the pdf and did not find any font substitution. May be the problem comes from the referee printer?

**SPECIFIC COMMENTS:**

(1) For this assessment of time series of iceberg properties, assessment of uncertainty are important and should be included. In particular, some of the quantities in Figures 4 and 5 should be plotted with uncertainties. Others might be OK with uncertainties indicated in the figure caption.

We introduces the uncertainties estimates provided in the previous study (Tournadre et al 2015). We choose not to include the uncertainties in the plot because it overloads the figure.

TCD
(2) The summary raises some interesting points about possible overestimates in previous studies of freshwater flux due to icebergs. But I haven't found a clear estimate of freshwater flux in this study. Is it possible to provide a concrete number?

**We provide general estimates in the text.**

(3) Mathematical notation is not type-set with embedded fonts. My print out has "square root" symbols in place of "less than or equal" and "greater than or equal" signs for the inequalities on line 6 of p. 2. Likewise, vectors in equation (3 have lost their arrowheads in my printout. For a journal that people read from downloaded pdfs, this is especially challenging. I admit that the cryosphere models are outside my domain of expertise, and I have trusted the authors on these and have not attempted to work through the background literature to evaluate their appropriateness.

I think the problem comes from the reviewer's printer or computer. We check carefully the pdf on the TC website and did not find any problem with the math and equations.

**DETAILED COMMENTS:**

(4) References are not cited correctly in many places, and a number of references that should be in the main text have been embedded in parentheses. This needs to be proofread carefully for citation style.

We check the reference and remove unnecessary parentheses.

(5) The writing, while generally OK, includes passages that are difficult to understand. I cannot possibly identify everything, and I encourage the authors to locate a native English speaker who can help them proofread in detail.

We has the manuscript proofread by a Canadian colleague.

(a) Here is my suggested rewriting of the abstract:

The evolution of the thickness and area of two large Southern Ocean icebergs that have drifted in open water for more than a year is estimated through the combined
analysis of altimeter data and visible satellite images. The observed thickness evolution is compared with iceberg melting predictions from two commonly used melting formulations, allowing us to test their validity of large icebergs. The first formulation, based on a fluid dynamics approach, tends to underestimate basal melt rates, while the second formulation, which considers the thermodynamic budget, appears more consistent with observations. Fragmentation leads melting as the major process responsible for the decay of large icebergs. Despite its importance, fragmentation remains poorly documented. The correlation between the observed volume loss of our two icebergs and environmental parameters highlights factors most likely to promote fragmentation. Using this information, a bulk model of fragmentation is established that depends on ocean temperature and iceberg velocity. The model is effective at reproducing observed volume variations. The size distribution of the calved pieces is estimated using both altimeter data and visible images and is found to be consistent with previous results and typical of brittle fragmentation processes. These results are valuable in accounting for the freshwater flux constrained by large icebergs in models.

**Changed. Thank you for the time taken to rewrite the abstract.**

(b) p. 1, lines 17 and 18. What is meant by "buffer"? What is meant by "diffuse"? Should it be "diffusive"?

**Changed to reservoir**

(c) Language is in places too informal, with the use of contractions. For example, line 9 of p. 2: "can't" should be "cannot" in formal writing.

**Corrected**

(d) p. 2m lines 23-25. Change wording and punctuation to clarify: " identified three styles of calving during the drift: "rift calving", which corresponds to the calving of large daughter icebergs by fracturing along preexisting flaws; "edge wasting", the calving of numerous small edge-parallel, sliver-shaped small icebergs; and "rapid disintegration",
which is characterised by the rapid calving of numerous icebergs."

**Changed**

(e) p. 2, line 34. "allow to" -> "allow us to"; also p. 9, line 20. (It's a transitive verb.) Likewise for p. 4, line 19. "enables" -> "enables us"

**Changed**

(f) "ones" should not be used as a substitute noun in a comparison. For example, on p. 4, just before heading 2.2, you can say "For example, B17a was sampled by 152 altimeter passes during its drift and C19a by 258 passes." The word "ones" is unclear and non-standard English. Another example on p. 9, line 17: "measured one"->"measured loss".

**Changed**

(g) p. 9, line 3. Does it take several years for the iceberg surface temperature to depend on the ablation rate? Or should this say, "Icebergs can sometimes float for several years. After initial adjustment, the iceberg surface temperature depends on the ablation rate."

changed to "Icebergs can sometimes drift for several years. During its travel the iceberg's surface temperature will depend on the ablation rate. "

(h) p. 1, line 16. Rewrite to say, "However, their melting accounts for less than 20% of their mass loss, and the majority of ice loss (80%) is achieved through breaking into smaller icebergs (Tournadre et al., 2016)."

**Changed**

(i) p. 1, line 21. The word "between" doesn't seem clear hear. Perhaps the authors mean, "differ in their basal ice-shelf and iceberg melting" or "achieve different relative balances of basal ice-shelf and iceberg melting"?
Changed to "Global ocean models including iceberg components show that basal iceshelf and iceberg melting have different effects on the ocean circulation."

(j) p. 2, line 13. "law" -> "laws" (k) p. 3, line 10. Capitalize "Southern Ocean"

**Changed**

(I) p. 3, line 11. Not clear if this is a singular or plural noun. To clarify perhaps, "area, size, and shape have been estimated".

**Changed**

(m) p. 3, line 14. By saying "that drifted", this implies that there are a number of other icebergs that drifted for more than two years in the South Pacific. I think the meaning would be clearer if "that" were replaced with "and".

**Changed**

(n) p. 3, line 15. "a relatively small 200 km2 one drifting" ->"relatively small (200 km2) and drifted"

**Changed**

(o) p. 3, line 15. To clarify the distinction between the plums and the big icebergs, say "both large icebergs".

**Changed**

(p) p. 3, line 5(2nd case). "confronted to" -> "confronted with".

**Changed**

(q) p. 4, line 21. "small icebergs location" -> "small iceberg locations". (The word "iceberg" is used as an adjective, and nouns used as adjectives are nearly always singular.)

**Changed**

TCD
**(r) p. 4, line 8. "ones" -> "passes" (s) p. 4, line 20. "constrains" -> "constraints" (t) p. 4, line 18. "as iceberg" -> "as an iceberg"**

**Changed**

(u) p. 5, line 17. Try "For each image with good cloud clover and light conditions ...."

**Changed**

```
(v) p. 5, line 28. "proxy of" -> "proxy for"
```

**Changed**

```
(w) p. 9, line 4. Try "can theoretically warm up to ...." (x) p. 9, line 9. "shows" -> "show" .... "ones" -> "velocities"
```

**Changed**

```
(y) p. 9, line 10. "thus considered as" -> "treated as"
```

**Changed**

(z) p. 9, line 17. "one" -> "loss" (aa) p. 9, line 25. Try "The second model parameter Ti (see Figures 6-c and 7-c) varies between -20âUeC2 and -0.6âUeC2 for B17a, with a  $-10.9\pm7.1$ âUeC2 mean for B17a. For C19a, it is between -9âUeC2 and 1âUeC2, with a  $-10.6\pm5.8$ âUeC2, although the model sometimes fails to converge to realistic iceberg temperature, i.e. for Ti<0âUeC2. This occurs ...."

**Changed**

(bb) p. 9, line 34. "fail" -> "fails" (cc) p. 11, line 29 Try "calving of icebergs from glaciers or ice shelves"

**Changed**

(dd) p. 12, line 13. "exist" -> "exists"

**Changed**
(ee) p. 12, line 17. "We investigate this matter by progressively including the dependence on environmental parameters ...."

**Changed**

```
(ff) p. 12, line 23. "ones" -> "loss"
```

**Changed**

(gg) p. 13, line 13. Clearer wording perhaps: "tested but brought no improvement".

Changed

(hh) p. 13, line 35. Missing superscript for km2

**Changed**

```
(ii) p. 13, line 17. "fragments" -> "fragment"
```

**Changed**

(jj) p. 14, line 22. "in open ocean" -> "in the open ocean"

Changed

```
(kk) p. 14, line 22 "relatively" -> "are relatively"
```

Changed

(II) p. 14, line 24. "get"  $\rightarrow$  "obtain" (The word "get" sounds too colloquial for formal writing.)

**Changed**

(mm) p. 14, line 29 Try "the first is more dynamically based, and the second results from a thermodynamic balance"  $% \left( \frac{1}{2}\right) =0$

Changed

TCD
(nn) p. 15, line 7. "chose to carry" -> "carried", "find out which" -> "identify the", "parameters are more likely to" -> "parameters that likely favour"

**Changed**

(oo) p. 15, line 23 onward. I'm not sure what is meant by this discussion. Change to "small iceberg bias". Does the "them" in "To include them" refer to small or large icebergs? The context suggests large icebergs, but the wording implies small icebergs. The word "still" should be removed.

Changed to "As a consequence, it is believed that the current modelling strategies suffer from a "small iceberg bias". To include large icebergs in models requires to ascertain that the previous modelling strategies are still valid for large icebergs. We also ought to gain more knowledge on how these bigger bergs constrain a size transfer to produce medium to small pieces via fragmentation. Eventually, these smaller pieces are those that account for the effective fresh water flux in the ocean."

(pp) p. 15, line 5-7. I would remove the "On the one hand" /"On the other hand" structure. It's a bit informal and doesn't clarify the meaning. The second sentence can begin "It also has demonstrated ...."

Changed

**Anonymous Referee #2**

Received and published: 20 December 2017

This paper presents an assessment of two different melting model approaches for icebergs during their drift and introduces an empirical fragmentation law developed from observations for the fracturing processes during iceberg drift. In principle, this is an interesting story with potential for improving models of freshwater input into the Southern TCD
Ocean by iceberg melting. However, the manuscript needs some work before publishing. A main problem is a lack of structure, which makes the line of thought hard to follow for the reader.

The manuscript contains a lot of different types of observational data, models, model results, so that it might have been better to divide the story into two manuscripts.

Melting and fragmentation are closely linked and we think it is important to conduct a joint analysis of the two processes in the same paper.

The introduction is rather long and detailed, but at the same time is lacking a clear line of thought. It should be more concise, and more importantly make the contribution or the potential of the presented approach to larger research goals more clear.

The introduction might be a little bit too long but we think that it is important to give a precise general context and to show why it is important to better understand and model the melting and fragmentation of icebergs. If the referee has any suggestions as to improve the introduction we are willing to include them.

The rest of the manuscript does not follow the usual methods/ results/ discussions structure. After the introduction a "data" section follows, after which already the results from the iceberg observations are presented.

We think that we do follow the general/usual structure of method/results/discussion. Data are generally considered as an integral part of the "method" section as they are the base of the study. Section 3 that describes the evolution of the two icebergs can also be considered as part of the"method" section.

Then one melt model is introduced, and the results presented, before the second model approach is described. This is not good for the reading flow. My suggestion for a better structure would be to clearly divide the paper in two parts, 1. Melting, 2. Fragmentation, and follow a classic methods/ result / discussion structure in each of the separate parts. The introduction and an additional joint discussion then should make it clear how these

TCD
parts belong together. In the first part you could have a "methods" chapter where the remote sensing data and their analysis is described, and the two melt approaches as well as the explanation that you are assessing and comparing their performance. Then present the melt results from observations, and the model results. Followed by a discussion of all three results. In a second part the fragmentation model could be explained. In this way the paper could be made more concise and clearer.

We don't really see how this structure is really different from the one we used. The "method" section consists of the data set used an the description of the evolution of the two icebergs, i.e. the basis of the study. The melting and fragmentation are then analyzed in the two following sections. We introduce a discussion subsection in the melting and fragmentation sections.

The summary is too long and repetitive, and contains parts which should be mentioned before in a discussion. In my view it would be necessary to thoroughly rework the structure of the paper in order to communicate the actual value of the study.

**We shorten the summary.**

Language: There are problems with punctuation, grammar and expressions in some places, which need a revision and maybe a read-through by a native speaker.

Using referee 1 comments we did our best to correct the text. It was also proof-read by a native speaker.

Figures: The labels on the axes should appear as the same font size, and should not overlap as in figure 4. I would suggest including a table with the fitting parameters for the fragmentation law, instead of printing the equations into the figure. Where possible I would place the panel labels outside of the main plot area and without a frame.

**Changed**

Specific comments:

TCD
Title: the title is not an adequate description of the content, as previously established melt models are being assessed, and a new empirical law for fragmentation is presented. Southern Ocean is a name and should be written with capital letters.

It is true that the melting and fragmentation laws are not treated in the same way in the paper but we don't think the title is misleading.

Page 1, line16: "melting accounts for less than 20 % of their mass loss" This is only true for the final stages of decay, as most large tabular icebergs keep their shape quite well during drift. This expression is also a bit misleading, as in the end all mass is lost due to melting, as also the smaller icebergs do melt.

The Tournadre et al (2015) clearly shows that the melting of large icebergs that transport most of the ice volume is limited compared to the breaking during their all life-cycle. It is a truism to say that ultimately they all melt.

Page 3, line 13: please insert the web-address of the BYU data base as a reference.

We gives the BYU and NIC web-address in the Acknowledgments section.

Page 5, line 15: insert "Here," in front of the second sentence to make it clear that now you are talking about your study and no longer about the BYU data.

**Changed**

Page 5, line 37: "Due to lack of a better alternative. . .SST is used": I understand that the ocean temperature at the base of the iceberg, i.e. in about 300m depth, is not easy to obtain, but it would be necessary to at least discuss this as an error source and get some data from models to estimate the difference possible between the temperature at the surface and at depth.

We include a reference to a paper by Merino et al (2016) that analyzed the difference between surface and 0-150 m mean temperature. (Merino et al, "Antarctic icebergs melt over the Southern Ocean : Climatology and impact on sea ice", Ocean Modelling
**(2016), 99--110).**

Page 6, line 9: I think SI units are standard for TC.

cm are SI units .

Page 7, first paragraph: Something that is completely missing here is the influence of firn compaction or changes in density along drift. This can have a substantial effect on the freeboard of an iceberg, while no mass is lost. This should be explained and an error should be estimated for this. There is also no information about which mean density has been assumed and why.

The firn densification is discussed in details in &4 and in annex A. Section 3 describes the data and the observed melting of the icebergs.

The units in this and the following paragraphs are not displayed correctly in the pdf.

We checked the pdf on the TC web site and did not find any problem with the display of units.

Page 10, line 16: "melt rate"

Corrected

Page 10, last paragraph: here methods, results and discussion are all mixed up in one single paragraph.

See general reply

Page 11, line 18: "melt rate"

**Corrected**

Page 12, first paragraph: here also all in one paragraph: first discussion and interpretation of results ("highest correlation is obtained for. . .), and only after this a reference to the figure where the data is shown. This is confusing to read. First describe the results, then interpret and discuss them Interactive comment

Page 12, line 17-19: This paragraph is unclear, are there words missing? ("volume loss depending"?)

Changed to "We investigate this matter by step, by progressively including the dependence to environmental parameters in a simple model of bulk volume loss."

Page 13, first paragraph: As the model is derived by fitting the observations, it should not be a surprise that there is a good correlation.

99.8% is better than a good correlation and it is not obvious that the same model can be apply to both icebergs with such a high correlation.

It would have been interesting to discuss the meaning of the fit parameters, and why they are different for the two icebergs.

The parameters are not significantly different that is what authorize to define a common model between the 2 icebergs.

In my view a useful empirical model should be able to reproduce the fragmentation of different icebergs with the same parameters.

That's exactly what is demonstrated in this section: the same model reproduces well the fragmentation of both iceberg.

TCD

---

## Referee Report (RR1)

[referee-annotated manuscript omitted]

---

## Author Response (AR2)

**Reply to referee's comments**

13th June 2018

The authors wish to thank referee 1 who spent time to correct many grammar and spelling mistakes and to provide us with many useful comments.

**Anonymous Referee #1**

The revised manuscript is improved over the original submission.

From a science standpoint, this is an interesting study, which takes advantage of new satellite products to evaluate the mechanisms driving the breakup of iceberg s (basal melting and fragmentation). The study shows that existing models are r elatively successful at producing basal melting and fragmentation estimates that are consistent with observations. This is valuable information, which will prove useful in advancing modeling of freshwater inputs to the ocean from icebergs.

The manuscript still suffers from substantial English problems (despite the fact that the authors say that they had a Canadian colleague proofread the manuscript---next time they may need to find a colleague who is willing to make more heavy handed use of a red pen.) Because I found myself marking corrections throughout the manuscript, so I eventually just created an annotated pdf. Please see the attached document. I do not know if I was able to flag every error. I would strongly recommend that the authors run a spell checker and grammar checker if possible.

Thank you for your time we did our best to correct the typos and erros.

While it is easy for authors to blame font substitution problems on the reader, that is not a constructive way to communicate with the audience. Regardless, I did not find font substitution errors in this version, and I see that the pdf file indicates embedded fonts.

Sorry but we checked again the pdf (both initial and revised versions) and didn't find the equations errors signaled by the reviewer.

**Anonymous Referee #3**

Bouhier and co-authors present an in-depth study of the melting and fragmentation of two large Antarctic icebergs. The study is concerned with an important topic in climate and cryospheric physics; a topic which has seen a recent surge of interest. The ideas and methods underlying this study are a good fit for The Cryosphere. However, I have a few major and more minor concerns that I believe should be addressed before this manuscript is accepted for publication.

Major Comments:

- I am aware that the authors have had somebody proof-read the language of the manuscript. However, there are still a large number of grammatical errors, typos, and formatting issues in this revised version. This has made reading and reviewing the manuscript unnecessarily difficult. I want to illustrate this with just the first lines of the introduction:

1.16: misplaced superscript "3" 1.17: "(1.500 km3 yr-1 80%)" -> "(1500 km3 yr-1, 80%)"

1.17: "Tournadre et al. (2016)" -> "(Tournadre et al., 2016)" 1.18: "as a reservoir to transport ice" -> "as reservoirs transporting ice"

I.18: "Antarctic Coastline" -> "Antarctic coastline"

l.19: "diffuse" -> "diffusive"

1.20: "the ter input" -> "the water input"

Sorry for the typos and errors. We did our best to correct the typos and errors (many thanks to reviewer#1 for the corrected pdf)/

I could go on. I would keenly urge the authors to revise the language and format to bring it up to the high standard appropriate for The Cryosphere. I would advise to consult a native English speaker once more.

- P.11 Estimation of V\_w and T\_i: As far as I understand the method here, the authors use one equation (eq 3) to determine two unknowns (V\_w and T\_i). This system is thus underconstrained, no? Please explain your

We thought it was clear enough in the text that the solution to estimate two unknowns from one equation is the minimisation of the difference between model and observations as stated in the text. We changed the sentence to

As current velocities and iceberg temperatures are not constant during the iceberg's drift, the modelled thickness loss is fitted to the measured loss for each time step  $t_i$  over a  $\pm 20$ -day period by selecting the  $V_w(t_i)$  and  $T_i(t_i)$  that minimise the distance between model and observations.

- P.11, P.15, P.17: I'm confused about the "99.9% correlation" between the models and observations (and reviewer #2 has hinted at this, without a satisfactory answer, in my opinion). Since the models are fitted to the observations (over small time steps) isn't a high correlation guaranteed by design?

To fit a model doesn't guarantee a high correlation. Even if the model is inadequate, there is still a solution that minimises the distance between model and observation. This solution can have a low correlation with model. It is true that if the model is adequate the solution that minimises the distance will have a high correlation.

Or rather, can you speak of "correlation" in the typical sense here?

I don't understand. We use correlation in the mathematical sense of correlation coefficient.

I see this issue with all 3 models that are discussed. Furthermore, If I understand this correctly I would have to disagree with the first line of the discussion (p.15 l.25): the authors have merely fitted V\_w and T\_i (in an underconstrained way(?)) such that the modeled loss of thickness matches the observed. P.15 l.25 reads as if the model ran independently from the observations and recovered the same thickness evolution. This is certainly not the case.

We think that it is clearly stated in the text that we fit the models to the observations and that this method allows to reproduce the observed variations of thickness with high precision. May be the sentence was not clear enough, we changed paragraph 3.3

- On closer inspection it becomes clear that the two models of eq (3) and eq (5) are not that unlike each other. Both depend (slightly non-linearly) on relative velocities and linearly on the relative temperature difference between ice and water. However, a direct comparison between the two models is made difficult by the different notations used. The models should be formulated as similarly as possible to make a comparison more intuitive.

The two parameterisations of the melt rate differs primarily in their representation of the heat transfer coefficient  $\gamma_T$ . The Week and Campbell parameterisation can be considered as a bulk and a simplified version of the Hellmer and Olbers one. The notation we use is the one used in all the literature and are identical for both parameterisation. We could provide an annex presenting a theoretical comparison of the two parameterisations. However, we think that it would be, firstly, quite long (at least 3 pages) and, secondly, that it won't be of great interest for the modeling community.

Also, there are some issues with units (e.g. unit of the 0.58 prefactor in (3), unit of water viscosity (p.12, 1.25)?).

**We introduce the units for both parameters.**

It would be informative to see how the two models compare for standard values of the drag and material coefficients. I'd suggest a plot for M\_b as functions of V\_w-V\_i for both models (although V\_w is presumably a different velocity in eq (5), or as functions of T\_w-T\_i (or T\_b-T\_w).

Equation 3 shows that Mb depends on the iceberg's longer axis (L) maximum length, the temperature difference and the velocity difference while the Equation 5 shows a dependency on velocity difference (through u\* and  $\gamma_T$ ), the temperature difference between the iceberg and the boundary layer and the temperature difference between the boundary layer and the water. It is quite difficult to make a significant plot as the parameters are quite different. In figure 1 we plotted the ratio of the thermal turbulent melting rate and forced convection one for L=120 km (top) and L=30 km (bottom) as a function of water temperature and velocity difference.  $T_i$  is fixed to  $4^{\circ}C$  and  $T_b$  to  $-2^{\circ}C$ . The ratio depends on the iceberg's length. It is of the order of 5 for icebergs around 30-50km and velocity difference

Figure 1: Ratio of equation 5 over equation 3 melting rates for L=120 km (a) and L=30 km (b) as a function of water temperature and velocity difference.

We don't understand the remark. Why 4.2 and not 4.1. The two paragraphs briefly describe the two melting parameterisations.

- Regarding firn compaction: I agree that it is important to mention this in the main text and to provide the 2-5% error estimate. However, I would argue that it doesn't need to get a full appendix (the error is small and the matter is rather tangential to the story). I thus recommend just removing Appendix A.

It was a demand from reviewer 2.

Minor Comments:

P.3

1.13: "area, size, and shape" - What's the distinction between area and size here? Does size refer to longest horizontal dimension(s)? Please clarify.

Changed to area

l.21: "The first section" - The Introduction is really the first section. You should probably refer to the sections by the numbers they are given.

Changed

Figure 1: - mark grounding sites - change time labels on legends to Jan 2014, Feb 2014, ...

Changed

P.4

L.6: delete "(latitude, longitude)" L.9: "Altimeter data can" L.17: "final detectable collapse" Changed

Figure 2: - add a legend with "\* - MODIS, o - Altimeter"

Already in figure caption

P.5 Section header: "2.3 Environmental data"

Changed

P.7

L.14 So the +- 0.9m represents the standard deviation of the standard deviation? I would just report the std as +- 3m. Or am I misunderstanding?

Correct. Changed to +-3m.

L.18 It's difficult to reconcile these numbers with Fig. 4a. There seems to be a faster melt period between Sept '14 and Nov '14? The melt appears to be slowing down again after May '15?. If you want to give these three regimes you should probably indicate the slopes with dashed lines?

The numbers in the text are complementary information and help understand the figure. Adding slopes with dashed lines won't give much information and will crowd the figure.

L.23 Stern et al (2016), "Wind-driven upwelling around grounded tabular icebergs" talks particularly about the unbalanced forces around grounded icebergs.

Added ref.

Last paragraph: If I understand this calculation correctly it assumes that all sidewall erosion is due to fracture and all bottom erosion is due to melt. I agree that this is a good approximation, but it should be stated explicitly.

Correct. We added, For large icebergs, the sidewall erosion/melting, which is of the order of some meters per day, can be considered as negligible compared to breaking.

Equation 1: This has a dimensional issue. The right hand side is  $M = dV/dt = m^3.d^{-1}$ ? The l.h.s is  $m^2*m$ . I guess you assume dT has units m.d^-1. You should probably write something like M = Delta V/Delta t = S\* Delta T/Delta t, where Delta t = 1 day. I'd argue for the use of Delta T, rather than dT, since you're looking at finite intervals.

The text was not clear enough and allowed a confusion between M the cumulative volume loss by melting and  $M_b$ the melting rate. Here M is in m3 and dT in m. We changed the sentence and the equations for M and B.

Equation 2: Similar arguments as for eq (1)

idem

Figure 4 - panel a. The caption doesn't match the colors of the figure. Also, maybe make the stars the same color as the continuous lines (red and blue?)

We changed the lines and the individual measurements (circles and stars) and the figure caption for figures 4 and 5.

Equation 4: what do the different terms represent physically?

We think that the sentence introducing the equation clearly describes the different terms. "It assumes heat balance at the iceberg-water interface and was originally formulated for estimating ice shelf melting. The turbulent heat exchange is thus consumed by melting and the conductive heat flow through the ice:"

P.16

L.20 delete parentheses

Figure 2: Comparison of the cumulative relative volume loss by fragmentation for B17a and C19a.

Done

L.23: what are the 63% and 64% values? Correlation coefficient r? Changed to 0.63 an 0.64. We changed all correlation from % to linear. P.17 L.1: "a-dimensional loss" -> "relative volume loss"

changed

Equation 8: While the form of eq (7) makes obvious sense, I don't have an intuition of why a second dependence on V\_i should be of the multiplying form  $(1+\exp(...V))$ . Could the authors explain this choice?

As shown by the correlation analysis  $M_{fr}$  depends primarily on temperature and secondly on velocity. To introduce a second order dependency on velocity we consider the velocity contribution as a corrective term so in the form of 1 + correction

Figure 8. I find it hard to see anything in this figure. The panels should be substantially revised and rethought. To start out with, I suggest two columns, with column 1 for C19a and column 2 for B17a.

The main point of this figure is to show the correlation between DV/V and environmental parameters. As it is we think it clearly shows the primary correlation with SST and the secondary one with velocity. We don't think that doubling the number of subplots would improve the comprehension.

Figure 9. I would put these on a log-lin scale (by construction of equations (7) and (8) this seems, no?). We changed the y scale of the plot to log.

Furthermore, I'd suggest a plot where the B17a and C19a curves are laid on top of each other to compare the two melt rates visually.

The comparison is not very significant as the two icebergs experienced very different environmental conditions especially near the end of their drift. We don't think that such a figure will be real interest for the study.

Fig 11. Some of the labels appear to be messed up, although I'm not entirely sure which ones Corrected. There was an inversion of the x and Y label for subplot a.

---

## Author Response (AR3)

**Reply to editor's comments**

21st June 2018

The title is not very specific and too generic.

changed to Melting and fragmentation laws from the evolution of two large Southern Ocean icebergs estimated from satellite data

The language can still be improved. For example:

"Fragmentation leads melting as the major process responsible for the decay of large icebergs."

Could this be rewritten as "Fragmentation is more important than melting for the decay of large icebergs."?

Although this sentence is grammatically and syntactically correct, it was may be to complex so we changed it.

Units: please check again all the units.

we re-checked the units using the Holland and Jenkins (1999) paper.

The dots in the units are strange. Should they mean middle dots?

changed to latex \cdot

Page 7: Equation 1: Explain meaning of i

i represents the ith day.

changed the basal melting volume loss $M$ is the sum of the products of iceberg surface, $S$ (in m$^3$), by the daily variation of thickness, $dT$ to

the basal melting volume loss $M$ at day $i$ is the sum of the products of iceberg surface, $S$ (in m$^3$), by the daily variation of thickness, $dT$

is S really in m^3 ? This should be a surface in m^2

Changed

Equation 3:

This is an empirical equation, therefore units shall be mentioned for every variable. The constant C does not agree with the constant given in Wagner et al. (2017) and the units seem unreasonable.

The Wagner value is $c = 6.7 \cdot 10^{-6} m^{-2/5} s^{-1/5} \circ C^{-1}$ , and ours is $C = 0.58 \mathrm{K}^{-1} \mathrm{m}^{0.4} \mathrm{s}^{0.8} \mathrm{day}^{-1}$ using $1 day = 86400 s$ $C = 0.58/86400 \mathrm{K}^{-1} \mathrm{m}^{0.4} \mathrm{s}^{0.8} \mathrm{s}^{-1} = 6.71 \cdot 10^{-6} K^{-1} s^{-.2} m^{0.4}$. The value is identical but there is an error in the Wagner formula on the sign of the power of meters -2/5 instead of 2/5 .

Equation 5: Units for salinity and pressure? For units of salinity see http://www.teos-10.org/

We added the units and changed psu to g/kg.

Tabel 1: use \exp instead of exp to avoid italics.

changed

Figure 13: there are some small dots at the right-hand side of the plot. Are these real data points?

No it was an artifact we redraw the fig.

Why are there lines connecting the data points?

The dashed straight lines represent the power law fit to the data. We changed the captions to include this.

What causes the gaps?

The pdf is computed over bins. For larger icebergs there are bins for which the number of iceberg is 0 causing a gap in the plot.

[revised manuscript text omitted]